# Quantitative sensing and signalling of single-stranded DNA during the DNA damage response

Susanne C.S. Bantele [1], Michael Lisby [2] & Boris Pfander [1]

The DNA damage checkpoint senses the presence of DNA lesions and controls the cellular response thereto. A crucial DNA damage signal is single-stranded DNA (ssDNA), which is frequently found at sites of DNA damage and recruits the sensor checkpoint kinase Mec1-Ddc2. However, how this signal – and therefore the cell's DNA damage load – is quantified, is poorly understood. Here, we use genetic manipulation of DNA end resection to induce quantitatively different ssDNA signals at a site-specific double strand break in budding yeast and identify two distinct signalling circuits within the checkpoint. The local checkpoint signalling circuit leading to γH2A phosphorylation is unresponsive to increased amounts of ssDNA, while the global checkpoint signalling circuit, which triggers Rad53 activation, integrates the ssDNA signal quantitatively. The global checkpoint signal critically depends on the 9-1-1 and its downstream acting signalling axis, suggesting that ssDNA quantification depends on at least two sensor complexes.

[1] Max Planck Institute of Biochemistry, DNA Replication and Genome Integrity, 82152 Martinsried, Germany. [2] Department of Biology, University of Copenhagen, DK-2200 Copenhagen, Denmark. Correspondence and requests for materials should be addressed to B.P. (email: bpfander@biochem.mpg.de)

DNA damage elicits a signalling response termed the DNA damage checkpoint. Once activated, the checkpoint induces several global (cell-wide) changes to cell physiology, including cell cycle arrest, transcriptional up-regulation of DNA repair genes and modulation of DNA replication pathways[1–4]. Furthermore, the checkpoint locally controls DNA repair[5,6].

Sensing of DNA damage occurs by the so-called apical or sensor kinases, which are recruited to specific DNA structures arising at DNA lesions. Budding yeast has two apical kinases: Mec1–Ddc2 (orthologues of human ATR-ATRIP) and Tel1 (orthologue of human ATM). Tel1 recognizes DNA double-strand breaks (DSBs) by interaction with the DSB-binding Mre11-Rad50-Xrs2 complex[7–9], while Mec1–Ddc2 senses the presence of single-stranded DNA (ssDNA) via interaction with replication protein A (RPA)[10,11]. ssDNA can be readily found at many lesion sites due to damage processing (for example, DNA end resection) or stalling of replication forks[12,13]. In fact, in budding yeast, the response to DSBs is dominated by Mec1–Ddc2 due to very active resection[14]. Upon sensing of the damage site, the apical kinases trigger a phosphorylation cascade, which leads to activation of downstream acting factors. Among them are the Rad53 and Chk1 effector kinases, which mediate cell-wide responses[4,15], or histone H2A, which upon phosphorylation forms the γH2A mark of damaged chromatin[16,17]. In this context, the apical checkpoint kinases face two tasks. On the one hand, they directly phosphorylate factors in the vicinity of the lesion site and thereby control the local response. On the other hand, they facilitate activation of the effector kinases, which subsequently localize throughout the entire nucleus and even into the cytoplasm[18] and phosphorylate checkpoint effectors. Consequently, apical kinases act upstream to set off the global DNA damage response.

Additionally, so-called mediators are required for checkpoint activation. Among these, the Rad9-Hus1-Rad1 (9-1-1) complex is loaded at the border of the ssDNA region (single-stranded–double stranded DNA (ss–dsDNA) junction) by the Rad24-RFC clamp loader complex in a manner that appears independent of Mec1–Ddc2 association[18–21]. The 9-1-1 complex serves as a platform for the association of additional checkpoint mediators (the 9-1-1 axis), such as Dpb11 (TOPBP1 in humans) and Rad9 (53BP1 in humans), which are critically required for recruitment, phosphorylation and activation of the effector kinase Rad53[22–28]. Notably, the checkpoint can become artificially activated even in the absence of DNA damage, if Mec1–Ddc2 and the 9-1-1 complex are forced to colocalize on chromatin, suggesting a sensor/co-sensor relationship[29].

It is logical to assume that the checkpoint not only qualitatively senses the presence of DNA lesions, but that quantitative signalling inputs are utilized to shape the cellular response to DNA damage. A highly quantitative signal integration is necessary, given the abundant occurrence of DNA lesions (with estimates ranging to up to 100,000 lesions per day in a human cell[30,31]). Most likely, cells are never entirely free of DNA lesions and thus require a dose-dependent response with a defined threshold of a tolerable DNA damage load. However, currently we do not understand how DNA damage signals are quantified.

Here, we investigate how the checkpoint quantifies the ssDNA signal at DNA damage sites. To this end we utilized a system of an enzyme-induced DSB in budding yeast[32], which allowed us to modulate the amount of ssDNA formed at a DSB using genetic manipulation of the DNA end resection process. Intriguingly, we find that specific checkpoint targets respond differently to quantitatively different ssDNA signals. Local γH2A phosphorylation appears hypersensitive and full-blown already at low levels of ssDNA, but unresponsive to further increases in the ssDNA

signal. In contrast, activation of the Rad53 effector kinase responds strongly to changes in the ssDNA signal. Quantitative signal transduction appears to depend on at least two sensors, as we observe that the association of not only the Mec1–Ddc2 kinase but also the 9-1-1 complex and its downstream factors are influenced by DNA end resection. Notably, we find that artificial hyper-activation of the 9-1-1 axis triggers hyper-activation of the Rad53 effector kinase. This occurs even under conditions of reduced Mec1–Ddc2 association, suggesting that the 9-1-1 signalling axis is a bottleneck for quantitative transduction of the ssDNA signal.

## Results

**γH2A and Rad53 respond differentially to changing ssDNA signals.** Single-stranded DNA is a universal DNA damage signal[1–3]. To investigate how the ssDNA signal is quantified, we studied the checkpoint response to a single site-specific DSB. At DSBs, 3' ssDNA is generated by DNA end resection, the processive nucleolytic digestion of the 5' strand[33]. Formation of ssDNA is an active process, which therefore allows genetic manipulation of the ssDNA signal using DNA end resection mutants. In order to induce a site-specific DSB at the MAT locus, we used galactose-induced expression of the HO endonuclease[32]. In M phase-arrested cells, DSB induction resulted in processive DNA end resection that reached up to 20 kb distal of the DSB in a 4 h timecourse, as visualized by chromatin immunoprecipitation (ChIP) against RPA (Fig. 1a, Supplementary Fig. 1A). In contrast, *exo1Δ sgs1Δ* cells deficient in long-range resection restricted ssDNA formation to less than 1.1 kb (Fig. 1a, see also refs. [34–36]). Mec1–Ddc2 directly interacts with RPA[10]. Consistently, in *exo1Δ sgs1Δ* cells, Mec1–Ddc2 association (visualized as the Ddc2-3FLAG ChIP signal) with the DNA damage site was strongly reduced and correlated with the amount of the ssDNA signal (Fig. 1a). Due to its direct interaction with RPA[10], Mec1–Ddc2 is the most intuitive candidate for a quantitative sensor of the ssDNA signal. Therefore, we expected the checkpoint response to be diminished in resection-defective *exo1Δ sgs1Δ* cells. Indeed, *exo1Δ sgs1Δ* cells showed much reduced recruitment of Rad53 to the break site and failed to phosphorylate and activate the Rad53 effector kinase over the timecourse of our experiment (Fig. 1a, b, see also refs. [36,37]).

In contrast, when we looked at a second Mec1–Ddc2 phosphorylation target—histone H2A[16,17]—using ChIP with an antibody specific for the γH2A mark, we surprisingly observed highly similar induction of γH2A phosphorylation in wild-type (WT) and *exo1Δ sgs1Δ* cells (Fig. 1a). This suggests that γH2A phosphorylation is hypersensitive and not quantitatively responding to a further increase of the ssDNA signal. The only clear difference in γH2A formation was observed close to the DSB (up to 7.6 kb for the 4 h timepoint, Fig. 1a, Supplementary Fig. 1B for an overlay), where the γH2A ChIP signal was consistently lower in WT than *exo1Δ sgs1Δ* cells. Given that RPA and γH2A ChIP signals appear anti-correlated, we suggest that this reduction in the γH2A signal occurs due to loss of histones on resected DNA. Otherwise, the γH2A ChIP signals (50–100 kb of DNA to both sides of the DSB) were remarkably similar in WT and *exo1Δ sgs1Δ*, independently of the availability of a repair template (Fig. 1c, Supplementary Fig. 1C) and independently of whether we measured γH2A distribution using either ChIP-quantitative PCR (qPCR) (Supplementary Fig. 1D) or over the entire damaged chromosome using ChIP-sequencing (seq) (Fig. 1c, Supplementary Fig. 2A–C).

Since the γH2A response requires less ssDNA signal compared to Rad53 activation, one might predict that it is also faster. Even though the temporal resolution of our experiments is limited

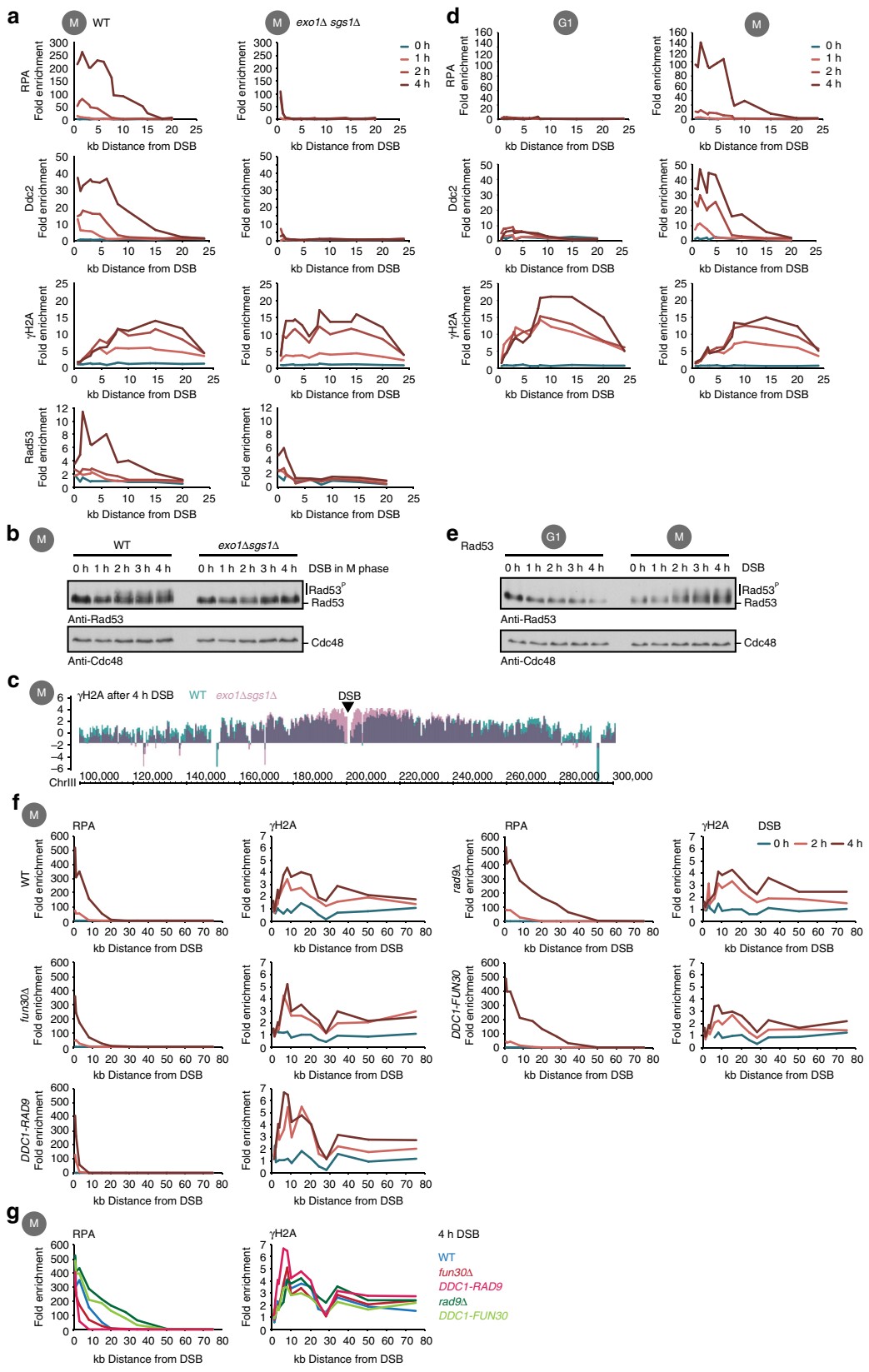

(30 min until a DSB is induced in most cells), we find that the γH2A response reached a plateau by 2 h of DSB induction, while Rad53 association with the damage site increased over the 4 h timecourse of the experiment (Fig. 1a).

To ensure that the observed effects were due to changes in the ssDNA signal, we employed cell cycle arrest as an alternative means to manipulate resection. Consistent with DNA end resection being highly cell cycle regulated[38], we observed very little RPA at the DNA damage site in WT cells that were arrested in G1, consistent with a strongly reduced ssDNA signal (Fig. 1d). Accordingly, Mec1–Ddc2 association and Rad53 activation were impaired in G1-arrested cells, as has been observed before

**Fig. 1** Differential regulation of DNA damage checkpoint effectors γH2A and Rad53. **a** Different amounts of Mec1–Ddc2 kinase phosphorylate H2A with similar efficiency. Wild-type (WT) and long-range resection-deficient exo1Δ sgs1Δ strains were arrested in M phase by nocodazole treatment, a non-repairable double-strand break (DSB) at MAT was induced by Gal-HO expression and protein recruitment was measured by chromatin immunoprecipitation (ChIP) at indicated times. Upper panel: fold enrichment of a given locus in a replication protein A (RPA) ChIP relative to undamaged control loci. Second panel: Mec1–Ddc2 kinase recruitment detect by ChIP against Ddc2–3FLAG. Third panel: H2A-S129 phosphorylation (γH2A phosphorylation). Lower panel: Rad53 kinase recruitment. **b** The checkpoint kinase Rad53 is activated in a resection-dependent manner. Western blot detecting the phosphorylation-dependent shift of activated Rad53 with an anti-Rad53 antibody and an anti-Cdc48 loading control. The samples were obtained at indicated time points after Gal-HO induction in M phase-arrested cells. **c** γH2A phosphorylation around a DSB is long-range resection independent. Overlay of γH2A ChIP-seq profiles around a DSB at MAT 4 h after DSB induction in WT cells (blue) and exo1Δ sgs1Δ cells (purple). Enrichment plotted relative to the whole genome average. **d, e** γH2A phosphorylation induced by a single DSB is not influenced by the cell cycle, while Rad53 phosphorylation is. ChIPs from WT cells as in **a**, but cells were arrested either in G1 by alpha-factor treatment (left panels) or in M phase by nocodazole treatment (right panels). **e** Western blot analysis of Rad53 activation as in **b**, but with G1- and M phase-arrested cells. **f** The DNA end resection regulators Fun30 and Rad9 do not affect γH2A phosphorylation. A DSB in M phase was induced in WT cells, hyper-resecting strains (rad9Δ, DDC1-FUN30 fusion) or resection-inhibited strains (fun30Δ, DDC1-RAD9 fusion). Resection (left panel, ChIP against RPA) and γH2A phosphorylation (right panel) were measured at indicated time points. **g** Overlay of ChIP-quantitative PCR (qPCR) traces of RPA and γH2A after 4 h of DSB induction from **f** in WT cells (blue), hyper-resecting cells (green) and resection-inhibited cells (red)

(Fig. 1d, e[38]). In contrast, γH2A phosphorylation was induced to similar extent in G1- and M-arrested cells (Fig. 1d), suggesting that γH2A phosphorylation is unresponsive to changes of the strength of the ssDNA signal and the amount of Mec1–Ddc2 associated with the DNA lesion during the cell cycle. We also find a very similar pattern of γH2A phosphorylation, when a DSB is introduced at another genomic location (Chr. VI, close to ARS607, Supplementary Fig. 2d). Moreover, we see that Rtt107, which is a direct reader of the γH2A mark[39], associates with damaged chromatin in a DNA damage-dependent manner, but its recruitment was unresponsive to the ssDNA signal, similar to what we observed for γH2A (Supplementary Fig. 2E).

Finally, we manipulated DNA end resection using mutants of the resection agonist Fun30 and the resection antagonist Rad9[40–44]. Fun30 and Rad9 are recruited to DSBs in a manner that depends on interaction with Dpb11–Ddc1 (subunit of the 9-1-1 complex)[23,40,45], and we have previously shown that covalent protein fusions with Dpb11 or Ddc1 can be used to artificially target Rad9 or Fun30 to DSBs and hyper-activate their respective function as resection regulators[40,46]. We find that the Ddc1-Fun30 fusion hyper-activated DNA end resection similar to a RAD9 deletion, whereas the Ddc1-Rad9 fusion blocked resection to an even greater extent than a FUN30 deletion (Fig. 1f, g). However, the damage-induced formation of γH2A was unchanged in these mutants (Fig. 1f, g).

Previous studies have shown that Mec1–Ddc2, and apical checkpoint kinases in general, have a dual function and act (i) in the local response at the lesion site and (ii) in the global, cell-wide response via activation of the effector kinases[4,15]. Our data collectively show that the two best-characterized outputs of these responses, phosphorylated Rad53 and γH2A, have fundamentally different dependencies of the ssDNA signal. We hypothesize that different signalling circuits lead to phosphorylation of Rad53 and to phosphorylation of H2A (see Supplementary Fig. 3 for a model). We will refer to the circuit leading to γH2A phosphorylation as local checkpoint circuit, since it is involved in controlling local action of repair factors. On the other hand, we will refer to the circuit leading to Rad53 phosphorylation as global checkpoint circuit, as it controls the cell-wide checkpoint response. Notably, our data indicate that already minimal ssDNA signals are able to elicit a full-blown local response that does not appear to correlate with the strength of the ssDNA signal. In contrast, the global response appears to feature a dose-dependent relation with the ssDNA signal.

**Determinants of the γH2A checkpoint signal.** Given that signalling in the local checkpoint circuit appeared to occur

independent of DNA end resection and the ssDNA signal, we tested other factors that might quantitatively determine γH2A phosphorylation. First, we tested whether H2A phosphorylation sites adjacent to the DSB were saturated. While we noted that the γH2A ChIP signal increased similarly over the HO-induction timecourse in different strains (Fig. 1a–f), arguing against saturation, we additionally addressed the question of saturation by reducing the density of H2A phosphorylation sites on chromatin. We made use of the fact that H2A is expressed from two gene copies (HTA1 and HTA2) in budding yeast and that both copies contribute similarly to the pool of H2A protein (1/3 and 2/3, see ref.[47]). We introduced the S129STOP mutation in either HTA1 or HTA2 to reduce the overall amount of H2A phosphorylation sites on chromatin (Fig. 2a) and controlled the effectiveness of our approach using the hta1-S129STOP hta2-S129STOP double mutant, which entirely abolished the γH2A ChIP signal (Fig. 2a, Supplementary Fig. 4A–B, note that IP/input ratios are plotted, since the hta1-S129STOP hta2-S129STOP strongly reduces the genome-wide γH2A background making normalization to control loci infeasible). Yet, γH2A phosphorylation after DSB induction was similar in WT, hta1-S129STOP and hta2-S129STOP cells (Fig. 2a, Supplementary Fig. 4A–B), suggesting that phosphorylation sites are not limiting.

Second, we tested the possibility that the two sensor kinases Mec1–Ddc2 and Tel1 might differentially contribute to γH2A phosphorylation in resection-proficient and -deficient conditions, as previous studies have indicated a switch from Tel1 to Mec1–Ddc2 during resection[16,18,48]. We observed in M phase-arrested cells that the DSB-induced γH2A ChIP signal was highly dependent on Mec1 (Fig. 2b, Supplementary Fig. 4C, note that MEC1 deletion also affected basal γH2A phosphorylation). Importantly, the same Mec1 dependence was also seen in long-range resection-deficient exo1Δ sgs1Δ cells (Fig. 2b). In contrast, we observed normal DSB-induced γH2A ChIP signals in M phase-arrested tel1Δ and tel1Δ exo1Δ sgs1Δ cells (Fig. 2c). Even in G1-arrested cells, we observed that mec1Δ sml1Δ cells showed a γH2A phosphorylation defect. In G1, however, tel1Δ showed a similar γH2A phosphorylation defect, and the DSB-induced γH2A ChIP signal was entirely abolished in mec1Δ tel1Δ sml1Δ cells (Supplementary Fig. 4D). Similar trends were also observed in an earlier study pioneering γH2A phosphorylation measurements by ChIP using a semi-quantitative PCR set-up[17] even though the tel1Δ defect was apparently less pronounced in our system. Furthermore, we also observed a minor role for Tel1 in γH2A phosphorylation in response to phleomycin-induced DNA breaks under conditions of resection inhibition in M phase (Supplementary Fig. 4E–G). Therefore, we conclude that at least

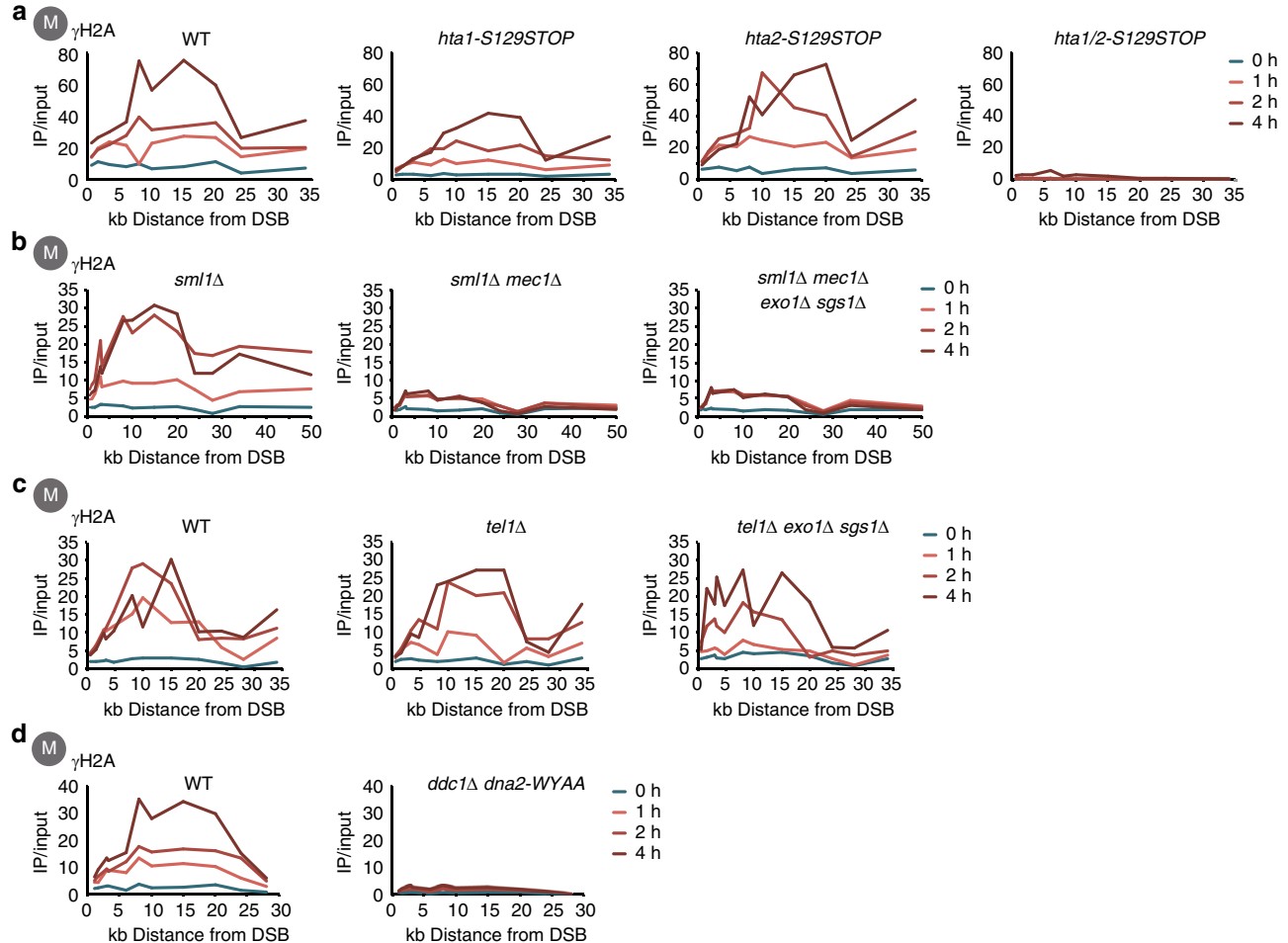

**Fig. 2** γH2A phosphorylation by Mec1 is not shaped by the H2A substrate or kinase redundancy. All data are plotted as IP/input ratios of γH2A chromatin immunoprecipitations (ChIPs). **a** The number of γH2A phosphorylation sites (H2A-S129) on chromatin is not limiting for γH2A phosphorylation efficiency. γH2A phosphorylation was measured in strains with either normal γH2A phosphorylation site availability (wild-type (WT)) or in strains in which the number of phosphorylation sites was reduced by mutation of either one or both H2A coding genes (*hta1-S129STOP, hta2-S129STOP*, respectively). **b**, **c** γH2A is mainly phosphorylated by Mec1-Ddc2 in M phase. ChIP-quantitative PCR (qPCR) analysis of γH2A phosphorylation around a double-strand break (DSB) in M phase cells. **b** WT, *mec1Δ sml1Δ* and *mec1Δ sml1Δ exo1Δ sgs1Δ* mutant strains were analysed at indicated time points. Deletion of the ribonucleotide reductase inhibitor Sml1 (suppressor of Mec1 lethality) confers viability in cells lacking Mec1 or Rad53. **c** WT, *tel1Δ* and *tel1Δ exo1Δ sgs1Δ* mutant strains arrested in M phase and analysed at indicated time points after DSB induction. **d** Mec1 activators are required for γH2A phosphorylation. ChIP-qPCR analysis of γH2A phosphorylation after DSB induction at MAT in WT cells and *dna2-WYAA ddc1Δ* mutant cells arrested in M phase

in M phase, γH2A phosphorylation at an HO-induced DSB is Mec1 dependent and that compensation by Tel1 is not causative for the apparently normal γH2A phosphorylation in response to a DSB in long-range resection-deficient cells. The contribution of Tel1 in G1 as judged by ChIP and the phleomycin experiment supports the Tel1-Mec1 switch model, but our data suggest that this switch occurs very early during resection even before the stage at which the *exo1Δ sgs1Δ* mutant is blocked. Overall, we conclude that very low amounts of Mec1–Ddc2 (and possibly also Tel1) are sufficient to induce full γH2A phosphorylation.

Third, we tested whether any of the established Mec1 activators[23,26,27,49,50] may be limiting to γH2A phosphorylation. We used a *dna2-WYAA ddc1Δ* strain to abolish Mec1 activation by either Dpb11, Ddc1 or Dna2[22,50,51] and found that in the absence of all three activators the damage-induced γH2A phosphorylation ChIP signal was largely abolished (Fig. 2d, Supplementary Fig. 5A–B). In contrast, we did not observe an essential role of any single activator in γH2A phosphorylation after DSBs, even though we observed that basal γH2A phosphorylation in M-phase cells was affected in the *dna2-*

*WYAA* strain (Supplementary Fig. 5C–E). Therefore, although γH2A phosphorylation is dependent on Mec1 activators, there is no specific activator that limits γH2A phosphorylation at an HO-induced DSB.

Lastly, we tested whether γH2A phosphorylation would be limited by a phosphatase. Previous studies have shown that PP4 is the major γH2A phosphatase[52,53] and we therefore tested γH2A phosphorylation in *pph3Δ* cells. Similar to the previous study, we observed that basal γH2A phosphorylation was increased in *pph3Δ* cells consistent with a role of PP4 in γH2A removal after completion of repair (Supplementary Fig. 5E–G). Moreover, γH2A phosphorylation was also increased after phleomycin treatment (Supplementary Fig. 5G), but it is unclear to what extent this result is confounded by a possible DNA repair defect or a potential accumulation of soluble γH2A[53,54]. In contrast, we did not observe a pronounced change of HO-induced γH2A (Supplementary Fig. 5E–F). We therefore conclude that neither the amount of kinase nor substrate or a specific kinase activator is limiting γH2A phosphorylation and that it is also not limited by PP4-dependent dephosphorylation.

Overall, our data are consistent with a model in which limited amounts of Mec1–Ddc2 are already sufficient to facilitate efficient γH2A phosphorylation. We furthermore note that H2A and Mec1–Ddc2 (Fig. 1a) are anchored at specific locations within the damaged chromosome. Thereby, enzyme substrate encounters are likely to dependent on chromosome architecture and perhaps mobility of these locations[55–60], which could pose a bottleneck to γH2A phosphorylation spreading.

**The 9-1-1 signalling axis responds to DNA end resection.** Mec1–Ddc2 association with the damaged chromosome mirrored the ssDNA signal and correlated with Rad53 phosphorylation (Fig. 1a, b). Thus, Mec1–Ddc2 levels at the DSB correlate with signalling in the global checkpoint circuit. Given the differential response of Mec1 targets to resection, we questioned whether other factors may contribute to the quantitative transduction of the ssDNA signal in the global checkpoint circuit. Notably, Rad53 phosphorylation in response to DSBs requires additional mediator proteins, namely the 9-1-1 complex (consisting of Ddc1, Mec3 and Rad17 in budding yeast) and the scaffold proteins Dpb11 and Rad9 (referred to as the 9-1-1 axis in the following). Previous data suggest that of all checkpoint mediators, the 9-1-1 complex is furthest upstream and facilitates recruitment of the downstream mediators in a manner that depends on phosphorylation of the C-terminal tail of Ddc1[18,22,23,51]. Our data are in agreement with this model, since *ddc1-T602A* cells lost the DSB association of Dpb11 or Rad9 and fail to activate Rad53 (Fig. 3a, b).

We next aimed to test whether the 9-1-1-axis was regulated by resection. To this end, we analysed DSB localization by ChIP and found that in *exo1Δ sgs1Δ* cells, association of all three factors was restricted to the immediate vicinity of the DSB, where resection occurred even in *exo1Δ sgs1Δ* (Fig. 3c). Likewise, in resection-deficient G1 cells we observed a reduction of DSB recruitment for the 9-1-1 subunits Ddc1 and Dpb11 (Supplementary Fig. 6A). These data indicate that DSB association of checkpoint mediators via the 9-1-1 axis is influenced by DNA end resection.

To obtain information on the number of 9-1-1 complexes associating with a DSB in single cells, we turned to quantitative fluorescence microscopy. In our system, induction of a single DSB allowed us to measure association of proteins as the appearance of a single, HO-dependent RPA-CFP and Ddc1-YFP focus (Fig. 3d, Supplementary Fig. 6B, C, see also refs. [18,21,61,62]). By comparing M phase- and G1 phase-arrested cells, we found that RPA focus formation coincides with resection (>50%, 4 h after HO induction in M-phase cells) and is strongly reduced in G1 phase cells (<12%, 4 h after HO induction), where resection is less active (Fig. 3e, Supplementary Fig. 6B–D).

To quantitatively determine the association of RPA and Ddc1 to the DSB we measured the fluorescence intensity of colocalizing Rfa1-CFP and Ddc1-YFP foci as an indicator of the number of recruited protein molecules. Furthermore, we used the fluorescence intensity of Rad52-CFP/YFP foci as a standard to deduce numbers of fluorescently labelled molecules in the focus[61]. Notably, we find that HO-induced foci accumulate Ddc1-YFP over time in M phase, as they do for RPA (Fig. 3f). Furthermore, we find a correlation ($R^2 = 0.37$) between the number of DSB-recruited 9-1-1 and RPA 4 h after induction of resection (Fig. 3g), suggesting that 9-1-1 complexes are continuously associating with chromatin during resection. We also note that the abundance of DSB-associated 9-1-1 complexes and RPA differs by at least an order of magnitude (Fig. 3f, Supplementary Fig. 6E). The DSB foci contain between 300 and 2000 molecules of RPA 4 h after DSB induction, consistent with resection rates of 4–5 kb/h[34] and an RPA footprint of 20/30 bases[63]. At the same time, they accumulate fewer than 30 9-1-1 molecules (Fig. 3f, Supplementary Fig. 6E).

Next, we analysed formation of RPA-CFP and Ddc1-YFP foci under mutant conditions that lead to either increased (*rad9Δ*) or decreased (*exo1Δ*) DNA end resection (Fig. 3h, see refs. [35,41,44,46]). Notably, we observed increased association of both RPA-CFP and Ddc1-YFP into HO-induced foci in *rad9Δ* cells (Fig. 3h) and conversely reduced association of RPA-CFP and Ddc1-YFP in *exo1Δ* cells (Fig. 3h). Notably, Ddc1–3FLAG ChIPs at a DSB site suggest a similar trend, with increased Ddc1–3FLAG and RPA ChIP signals in hyper-resecting *rad9Δ* cells and decreased Ddc1–3FLAG and RPA ChIP signals in hypo-resecting *exo1Δ* and *sgs1Δ* (Fig. 3i, Supplementary Fig. 6F). Altogether, these data suggest that the 9-1-1 complex associates in a resection-dependent manner with DSB sites, suggesting that it could be used as a second quantitative sensor that reports on the amount of DNA end resection. We caution, however, that 9-1-1 may also associate with DSBs in the absence of extended DNA end resection. Specifically, we observed surprisingly high Ddc1-YFP signals associated with the DSB in *exo1Δ sgs1Δ* cells, but notably these signals did not increase over time, suggesting that they were resection independent (Supplementary Fig. 6H–I, note that *exo1Δ sgs1Δ* have high sporadic DNA damage foci necessitating the use of a system of DSB induction where an I-SceI break site on chromosome V is marked by 336 TetO arrays, Supplementary Fig. 6Gii). These signals could correspond to strong Ddc1–3FLAG ChIP signals we measured in the immediate proximity of the DSB in *exo1Δ sgs1Δ* cells (Fig. 3c, Supplementary Fig. 6F). Overall, we therefore conclude that cells—when undergoing DNA end resection—show increased association of the 9-1-1 complex with the DSB over time. It seems therefore plausible that the quantitative nature of this association may be used as a proxy for the ssDNA signal in the global checkpoint circuit.

**The 9-1-1 axis critically contributes to the global checkpoint signal.** Since our data indicated that resection quantitatively controls the DSB association of not only Mec1–Ddc2, but also the 9-1-1 axis, we tested whether the amount of the 9-1-1 axis proteins at a DSB contributes to Rad53 activation in a quantitative manner. Therefore, we expressed a covalent fusion of the 9-1-1 complex with its downstream mediator Rad9 (Ddc1-Rad9 fusion). As expected, Rad9 recruitment to the DSB was markedly increased in the context of the fusion compared to WT cells (Fig. 4a). Moreover, as a consequence of the function of Rad9 as a resection inhibitor[44,46], resection was blocked and Mec1–Ddc2 recruitment was reduced (Figs. 1f–g and 4a). Intriguingly, and despite the decreased Mec1–Ddc2 recruitment, global checkpoint signalling as determined by Rad53 phosphorylation was elevated above WT levels (Fig. 4b). Enhanced signalling along the 9-1-1 axis therefore enhances global checkpoint activation, suggesting that the 9-1-1 axis is a bottleneck to checkpoint activation and that the 9-1-1 axis quantitatively contributes to global checkpoint signalling.

The Ddc1-Rad9 fusion bypasses the phosphorylation of the 9-1-1 complex by Mec1–Ddc2. Therefore, we utilized a second construct, in which Rad9 is fused to Dpb11 (Rad9-dpb11ΔN fusion,[23,46]) and which retains dependency on 9-1-1/Ddc1 phosphorylation by Mec1–Ddc2 (Supplementary Fig. 7A, B). Similar to the Ddc1-Rad9 fusion, also the Rad9-Dpb11 fusion lead to enhanced Rad9 recruitment to the DSB, while Mec1–Ddc2 recruitment and DNA end resection were decreased in its presence. Importantly, also in this genetic background, global checkpoint signalling was hyper-activated (Fig. 4c–d).

In summary, we discovered genetic conditions that enhance signalling specifically via the 9-1-1 axis. These conditions triggered hyper-activated global checkpoint signalling despite decreased recruitment of Mec1–Ddc2. This establishes a role of the 9-1-1 complex and its downstream mediators Dpb11 and Rad9—the 9-1-1 axis—as a bottleneck limiting Rad53 activation. Together, we suggest

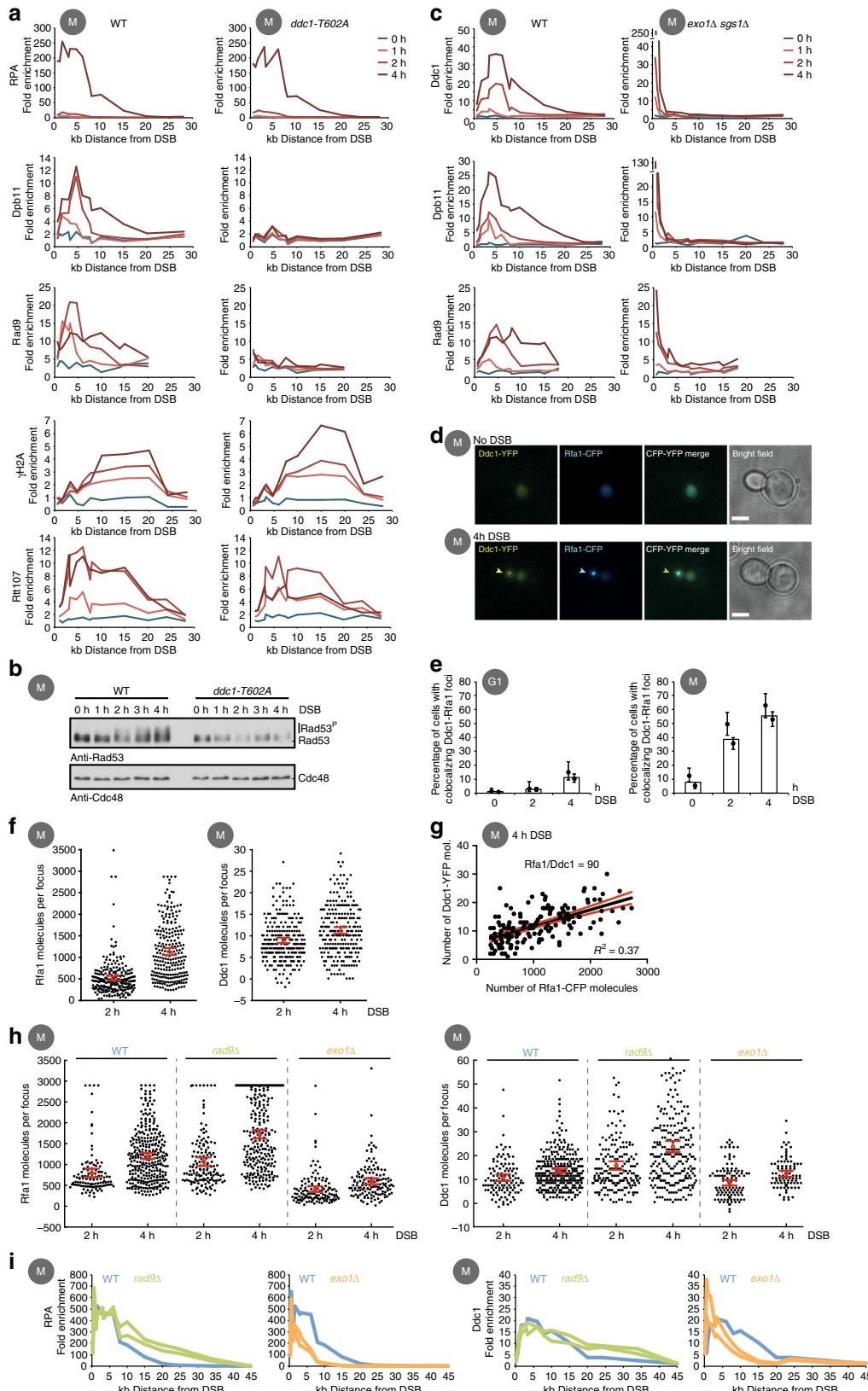

that the number of 9-1-1 molecules loaded at a DSB may be critical for a quantitative signal in the global checkpoint response.

## Discussion

DNA damage checkpoint signalling is shaped by a two-layered mechanism in the form of apical and effector kinases.

Interestingly, the apical kinases do not only transduce the checkpoint signal to the effector kinases, but also phosphorylate checkpoint targets on their own. Targets of apical and effector kinases can be distinguished by their localization. Apical kinase targets such as histone H2A, Rtt107 or the Smc5–6 complex act locally on the damaged chromosome and can be visualized as a

**Fig. 3** The Rad9-Hus1-Rad1 (9-1-1) axis is boosted upon ongoing DNA end resection. **a, b** 9-1-1 (Ddc1) phospho-site T602 is required for recruitment of Rad9, Dpb11 and Rad53. **a** Wild-type (WT) and *ddc1-T602A* cells in M phase were analysed by chromatin immunoprecipitation (ChIP)-quantitative PCR (qPCR) of indicated proteins after double-strand break (DSB) induction. **b** Analysis of Rad53 phosphorylation by western blot against Rad53 and Cdc48 as control in WT and *ddc1-T602A* cells. **c** Recruitment of Ddc1, Dpb11 and Rad9 is influenced by DNA end resection. ChIP-qPCR analysis against indicated proteins with WT (left) and *exo1Δ sgs1Δ* (right) cells in M phase at indicated time points. **d** Ddc1-YFP (yellow) and RPA-CFP (blue) form DSB-induced foci. CFP cyan fluorescent protein, RPA replication protein A, YFP yellow fluorescent protein. Representative microscopy images from indicated times after DSB induction in M phase. The scale bar represents 3 µm. **e** Ddc1 and RPA foci formation is most efficient in M phase. Percentage of cells with foci at indicated times after DSB induction from experiment in **d**. After 4 h, 55% of M phase-arrested cells but only 15% of cells in G1 show foci. Error bars indicate mean with 95% confidence intervals (G1 n = 587–818, M phase n = 489–693). **f** RPA and Ddc1 recruitment to a DSB in M phase increases over time. Scatter plot depicting the molecules per focus of Rfa1 (left) or Ddc1 (right) after 2 h and 4 h DSB. Same experiment as in **d, e**. **g** Correlation of RPA and Ddc1 recruitment to a DSB in individual cells. Scatter plot showing the number of Ddc1-YFP against Rfa1-CFP molecules per focus. Same experiment as (**d–f**). The black line represents a linear regression with corresponding 95% confidence intervals in red. **h, i** RPA and Ddc1 recruitment to a DSB are similarly influenced by resection. **h** Scatter plot depicting the number of Rfa1 (left plot) or Ddc1 (right plot) molecules per focus after 2 (left, respectively) and 4 h DSB (right, respectively) in WT, *exo1Δ* and *rad9Δ* cells. Error bars indicate 95% confidence intervals (n = 115–312). **i** ChIP-qPCR to measure Ddc1 (upper panel) and RPA (lower panel) after 4 h DSB in M phase comparing WT (blue) with *exo1Δ* (orange) or *rad9Δ* mutant cells (green)

focus surrounding the site of the DNA damage[16,64]. The effector kinases in contrast act cell-wide after a local activation step[4,18,65]. In other words, the apical checkpoint kinases trigger two distinct signalling circuits. Intriguingly, our study demonstrates that the DNA damage signal (in this case single-stranded DNA) is integrated differently by these two signalling circuits. The local checkpoint circuit leading to γH2A phosphorylation is hypersensitive and already fully active at low (<1.1 kb) ssDNA signals. In contrast, the global checkpoint circuit leading to phosphorylation and activation of the Rad53 effector kinase is able to quantitatively respond to a broad range of ssDNA signals and is not fully active even at >20 kb of ssDNA.

Our data are therefore consistent with a model, in which the DNA damage checkpoint is not a single pathway, but rather a composite of at least two distinct signalling circuits that can be discriminated by their ability to quantify DNA damage (Supplementary Fig. 3). It seems plausible that the local checkpoint response is sensitive to the presence of any DNA damage in order to control local repair, irrespective of the overall damage load. The global checkpoint response in contrast must accurately read the cellular DNA damage load in order to control cell-wide processes, such as cell cycle progression or DNA replication.

These findings raise two fundamental questions. First, by which mechanism is the γH2A phosphorylation defined if not inherently by the kinase recruitment? Second, if ssDNA-dependent phosphorylation events are not an intrinsic part of the Mec1 mechanism of action, which additional factors cooperate with Mec1–Ddc2 in order to relay quantitative signals?

For the local circuit we find that γH2A phosphorylation sites are not saturated under our experimental conditions (Fig. 2a), suggesting that a specific bottleneck is limiting H2A phosphorylation, which is neither the amount of damage-associated kinase nor the availability of phosphorylation sites. We furthermore ruled out that this bottleneck may be posed by a specific Mec1 activator. We therefore favour a model, whereby the bottleneck is formed by chromosome architecture and perhaps mobility. Given that substrate and kinase are tethered to specific chromosomal locations, contacts between these two chromosomal locations will determine the frequency of substrate kinase encounters[56–60]. Alternatively, activated Mec1–Ddc2 molecules may not be strictly tethered to the ssDNA stretch and could target substrates within a certain diffusion range. In both cases, however, chromosome architecture will influence how far the γH2A damage mark spreads into chromatin and will quantitatively shape the γH2A signal. Consistent with this idea, γH2A spreads in trans, guided by known inter-chromosomal contact points such as centromeres[66], and data from mammalian cells furthermore point towards restriction of the γH2A signal within topologically associated domains[55].

The global checkpoint signalling circuit leading to the activation of the Rad53 effector kinase is more complex. Most critically, it involves mediator proteins (9-1-1 complex, Dpb11 and Rad9; 9-1-1 signalling axis), which facilitate signal transduction to the effector kinase. Importantly, DNA damage recruitment of Mec1–Ddc2 and the 9-1-1 complex occur by separate mechanisms[10,20,21,29]. Therefore, the global DNA damage signalling circuit relies on two independent DNA damage sensors[21,29]. Qualitatively, the involvement of two sensors (or sensor and co-sensor) provides a fail-safe mechanism. Our data suggest, however, that additionally the involvement of two sensors is critical to quantify the ssDNA signal and to yield a proportional response.

How can a quantitative involvement of the 9-1-1 complex in the DNA damage checkpoint be explained? Our data point toward consecutive cycles of 9-1-1 loading during DNA end resection. Biochemical studies suggest that 9-1-1 is loaded at 5' ss–dsDNA junctions, the leading edge of DNA end resection[19,20]. However, RPA interacts strongly with the Rad24-RFC clamp loader complex[67] and formation of DSB-associated 9-1-1 foci depends on RPA[18]. Moreover, 9-1-1 loading occurs at the 5' ss–dsDNA junction and it is unclear how this loading is coordinated with the activity of resecting nucleases. Currently, we can therefore only speculate what happens to 9-1-1 complexes after loading. Potentially, loaded 9-1-1 complexes could slide ahead of the resection machinery on dsDNA or be left behind on resected DNA.

It appears highly likely that this mechanism of dual recognition of checkpoint signals is present in higher eukaryotes, too, given that the involved proteins as well as their association with single-stranded DNA are highly conserved throughout evolution. Notably, mammalian cells feature another activator for ATR (human Mec1), ETAA1, which binds to RPA as well, but is independent of the 9-1-1 complex[68–70]. It therefore seems reasonable to hypothesize that ETAA1 will—similar to the 9-1-1 complex—quantitatively contribute to checkpoint signalling.

We conclude that a main task of all upstream factors (apical kinase, co-sensor and scaffolds) in the global checkpoint circuit is to relay the DNA damage signal to the checkpoint effectors in a quantitative manner. This will allow cells to integrate the DNA damage load over the entire genome and tailor an appropriate cell-wide response. Such a mechanism appears essential given the abundance of endogenous DNA damage[30,31], where checkpoint signalling will typically arise at multiple DNA damage sites. Importantly, by generating a global checkpoint response that correlates with the ssDNA signal, different DNA lesions will contribute differentially to the overall checkpoint response, depending on how much ssDNA is formed. Moreover, this mechanism also sensitizes the global checkpoint response to S phase, where an abundant ssDNA signal can be formed through

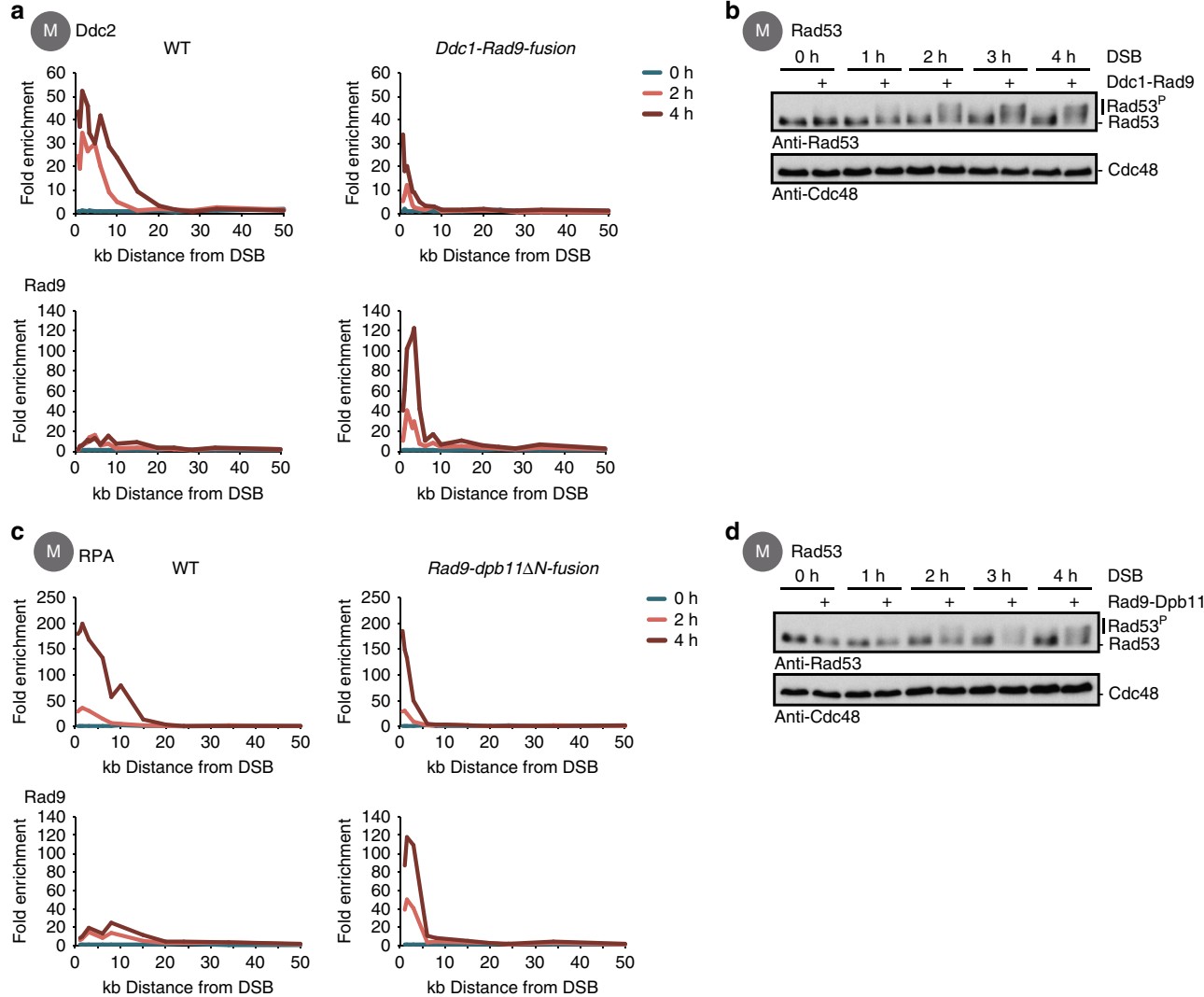

**Fig. 4** Hyper-activation of the 9-1-1 axis enhances Rad53 activation even at low levels of Mec1 recruitment. **a** A covalent Ddc1-Rad9 interaction results in enhanced Rad9 recruitment and blocks Mec1–Ddc2 kinase recruitment to a double-strand break (DSB). Wild-type (WT) cells and *DDC1-RAD9-fusion* cells (same construct as used in Fig. 1f–g) were arrested in M phase and analysed at indicated times. Depicted are chromatin immunoprecipitation (ChIP)-quantitative PCR (qPCR) measurements of Mec1–Ddc2 recruitment (upper panels, ChIP directed against Ddc2-3FLAG using an anti-FLAG antibody) and Rad9 recruitment to a DSB (lower panels, ChIP directed against Rad9 or the fusion which both carry a 3FLAG tag for detection). **b** Rad53 activation in response to a DSB is strongly enhanced in cells expressing a *DDC1-RAD9-fusion* construct. Western blot analysis of Rad53 activation at indicated times after DSB induction, strains as in **a**. Cdc48 served as loading control (lower panel). **c** A covalent Rad9-Dpb11 fusion protein enhances Rad9 recruitment to the DSB and blocks DNA end resection. WT cells and cells expressing *RAD9-dpb11ΔN* (lacking BRCT 1+2 of Dpb11 which normally bind to Rad9) were arrested in M phase and analysed at indicated time points. ChIP-qPCR measurements of DNA resection (replication protein A (RPA), upper panels) and Rad9 recruitment to a DSB (lower panels). To measure Rad9 recruitment, Rad9-dpb11ΔN-fusion and Rad9, respectively, were tagged C-terminally with a 3FLAG tag. **d** Rad53 activation in response to a DSB is enhanced in *RAD9-dpb11ΔN-fusion* cells. Western blot analysis of Rad53 activation at indicated times after DSB induction, strains as in **c**, Cdc48 serves as loading control

replication fork stalling, consistent with the essential function of Mec1–Ddc2 and Rad53 in S-phase regulation.

Lastly, we note that certain cell-wide responses such as cell cycle arrest or–in higher eukaryotes—apoptosis[30] are binary switches. This implies the existence of thresholds, above which a certain response is triggered. Indeed, also phosphorylation/activation of Rad53 does not show a strictly linear increase with an increasing ssDNA signal (Fig. 1a, b). We are only beginning to understand the quantitative nature of checkpoint signalling, but it will be critical to reveal how such thresholds are formed, how big a DNA damage load cells tolerate and whether these thresholds differ between organisms, cell types or during development. We think that this question is also of central relevance for our

understanding of cancer development, since the DNA damage checkpoint forms an important barrier that is often overcome by mutation during tumorigenesis[71,72].

## Methods

**Yeast strains and plasmids**. All yeast strains used in this study are derived from W303 MATa (strains listed in Supplementary Table 1) and were constructed using PCR-based tagging or deletion of yeast genes[73]. Cells were grown in YP glucose or YP-raffinose media at 30 °C. Cell cycle synchronization was performed using alpha-factor or nocodazole for 2–3 h and controlled by Flow cytometry.

For molecular cloning, genes were amplified from yeast genomic DNA and inserted in plasmids using the In-Fusion HD cloning kit (Clontech). For site-directed mutagenesis, a PCR-based protocol with mutagenic oligonucleotides was used. All plasmids used in this study are listed in Supplementary Table 2.

**ChIP and qPCR analysis.** For chromatin immunoprecipitation of γH2A, FLAG-tagged proteins and RPA, cells were grown in YP-Raffinose to an OD600 of 0.5 ($1 \times 10^7$ cells/ml) and—as indicated for the individual experiments–cell cycle arrest was induced. Ddc1, Dpb11, Rad9, Rtt107 and fusion proteins were tagged with a C-terminal 3FLAG tag, and ChIPs were directed against the FLAG tag. γH2A, RPA and Rad53 were pulled-down with antibodies directed against the respective protein. A double-strand break was introduced by inducing the HO endonuclease from the galactose promoter by addition of galactose to the cultures (2% final). The 100 ml samples were crosslinked with formaldehyde (final 1%) for 16 min at indicated time points and the reaction was quenched with glycine. Cells were harvested by centrifugation, washed in ice-cold phosphate-buffered saline and snap-frozen. For lysis, cell pellets were resuspended in 800 μl lysis buffer (50 mM HEPES KOH pH 7.5, 150 mM NaCl, 1 mM EDTA, 1% Triton X-100, 0.1% Na-deoxycolate, 0.1% SDS) and grounded with zirconia beads using a bead beating device. The chromatin was sonified to shear the DNA to a size of 200–500 bp. Subsequently, the extracts were cleared by centrifugation, 1% was taken as input sample and 40% were incubated with either anti-FLAG M2 magnetic beads (M8823, Sigma) for 2 h or 1.5 h with anti-RFA (AS07–214, Agrisera), anti-Rad53 (ab104232, Abcam) or anti-γH2A (ab15083, Abcam) antibody followed by 30 min with additional Dynabeads Protein A (10001D, Invitrogen, for RPA, Rad53 and γH2A ChIPs). The 2 μg antibody was used per sample. The beads were washed 3× in lysis buffer, 1× in lysis buffer with 500 mM NaCl, 1× in wash buffer (10 mM Tris-Cl pH 8.0, 0.25 M LiCl, 1 mM EDTA, 0.5% NP-40, 0.5% Na-deoxycholate) and 1× in TE pH 8.0. DNA-protein complexes were eluted in 1% SDS, proteins were removed with Proteinase K (3 h, 42 °C) and crosslinks were reversed (8 h or overnight, 65 °C). The DNA was subsequently purified using phenol–chloroform extraction and ethanol precipitation and quantified by quantitative PCR (Roche LightCycler480 System, KAPA SYBR FAST 2× qPCR Master Mix, KAPA Biosystems) at indicated positions with respect to the DNA double-strand break. A list of all qPCR primer sequences can be found in Supplementary Table 3. As control, 2–3 control regions on other, undamaged chromosomes were quantified.

**ChIP and sequencing analysis.** For the ChIP-seq experiment shown in Fig. 1c and Supplementary Fig. 2A–C, cells were treated as described above for the ChIP-qPCR experiments. Before de-crosslinking of eluted DNA-protein complexes, samples were digested with RNAse A. The sequencing library was prepared using the MicroPlex Library Preparation kit v2 (Diagenode) according to the manufacturer's manual. Size analysis and sequencing were performed by the genomics division of the LAFUGA lab (GeneCenter, Munich). The sequencing data were analysed in collaboration with Assa Yeroslawitz and plotted using the Integrative Genome Browser (IGB) software.

**Western blot analysis of γH2A and Rad53 activation.** For protein detection by western blot, $2 \times 10^7$ cells were harvested at the indicated time points and snap-frozen. Protein lysates were prepared by alkaline lysis and subsequent tri-chloroacetic acid precipitation. For analysis of γH2A, samples were run on pre-cast NuPage gels (4–12% Bis-Tris, Invitrogen) using MES buffer for 35 min at 200 V. To detect checkpoint activation by analysis of the Rad53 phosphorylation shift, samples were run on 10% sodium dodecyl sulfate-polyacrylamide gel electrophoresis (SDS-PAGE) gels for 180 min at 160 V. Western blotting was performed with standard methods. The γH2A phosphorylation was detected using anti-γH2A (ab15083, Abcam; 1:3000) antibody, for Rad53 shift detection anti-Rad53 (ab104232, Abcam; 1:4000) was used. As loading control, the membranes were washed and re-incubated with anti-Pgk1 antibody (22D5C8, Invitrogen; 1:7000) or anti-Cdc48 antibody (lab of Stefan Jentsch; 1:10,000). Unprocessed scans of all main figures and western blots can be found in the Source data file.

**Yeast live cell imaging.** Rfa1 was tagged with cyan fluorescent protein (CFP, clone W7) and Ddc1 with yellow fluorescent protein (YFP, clone 10 C)[18]. For live cell microscopy of Rfa1 and Ddc1 recruitment to an HO-induced DSB, cells were grown shaking in liquid SC+Ade medium (synthetic complete medium supplemented with 100 μg/ml adenine) with 2% raffinose at 25 °C to OD600 = 0.2–0.3 and arrested either in G1 phase with 10 μg/ml α-factor or in M phase with 15 μg/ml nocodazole for 2 h before addition of galactose to a final concentration of 2%. Cells were processed for fluorescence microscopy at the indicated times after addition of galactose as established[74]. Fluorophores were visualized on a Deltavision Elite microscope (Applied Precision, Inc) equipped with a 100× objective lens (Olympus U-PLAN S-APO, NA 1.4), a cooled Evolve 512 EMCCD camera (Photometrics, Japan) and an Insight solid-state illumination source (Applied Precision, Inc.). Images were acquired using softWoRx (Applied Precision, Inc.). Image analysis and fluorescence intensity quantification were done using Volocity (PerkinElmer) and presented as scatter plots using Prism (GraphPad software, Inc.) or Matlab (Mathworks). Images were pseudocoloured according to the approximate emission wavelength of the fluorophores.

**I-SceI induction.** Strains transformed with the I-SceI expression plasmid (pWJ1320)[61] were grown shaking at 25 °C in SC-Ade with 2% raffinose as a carbon source. To induce expression of the I-SceI endonuclease, galactose was added to the culture to a final concentration of 2% and incubation continued shaking at 25 °C.

**Repetitions of experiments.** All ChIP-qPCR experiments were performed in three technical replicates (qPCR).

The ChIP experiments in Fig. 1a were performed with different numbers of biological replicates, the RPA ChIP three times, the Ddc2 ChIP once, and the γH2A ChIP four times and the Rad53 ChIP twice. The ChIP-seq experiment shown in Fig. 1c was done once. ChIPs in Fig. 2d were performed with different numbers of biological replicates, the RPA and Ddc2 ChIPs three times and the Ddc2 ChIP twice. The experiment in Fig. 1f, g was performed in independent biological duplicates. The ChIP experiment in Fig. 2a was performed in two independent biological replicates for the double mutant and three replicates for the two single mutants. All other experiment shown in Fig. 2 were performed in two biological replicates. The ChIPs in Fig. 3a were performed independently, for each protein once. The ChIP experiment in Fig. 3c was repeated at least twice, or more often (Ddc1). The ChIP experiment shown in Fig. 3i was performed in two biological replicates. All ChIP experiments shown in Fig. 4 were independently performed twice. All western blot Rad53 activation experiments were repeated at least 5 times, and phleomycin experiments with a western blot read-out at least twice. All microscopy experiments shown in this manuscript were performed at least twice.

**Reporting summary.** Further information on experimental design is available in the Nature Research Reporting Summary linked to this article.

## Data availability

All data including yeast strains and plasmids are available from the authors upon reasonable request. ChIP-seq data are available from the National Center for Biotechnology Information (NCBI) Gene Expression Omnibus (GEO) under accession numbers: GSE124948; GSM3559929, IP sample WT 0 h RPA; GSM3559930, IP sample WT 4 h RPA; GSM3559931, IP sample exo1Δ sgs1Δ 0 h RPA; GSM3559932, IP sample exo1Δ sgs1Δ 4 h RPA; GSM3559933, IP sample WT 0 h γH2A; GSM3559934, IP sample WT 4 h γH2A; GSM3559935, IP sample exo1Δ sgs1Δ 0 h γH2A; GSM3559936, IP sample exo1Δ sgs1Δ 4 h γH2A; GSM3559937, Input DNA WT 0 h; GSM3559938, Input DNA WT 4 h; GSM3559939, Input DNA exo1Δ sgs1Δ 0 h; GSM3559940, Input DNA exo1Δ sgs1Δ 4 h.

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

## Acknowledgements

We thank Uschi Schkölziger for technical assistance, the sequencing unit of the Laboratory for Functional Genome Analysis (LAFUGA) at LMU Munich for next-generation sequencing, Assa Yeroslaviz and Tobias Straub for bioinformatic analysis of NGS data, Rohit Agarwal, Giovanni Cardone and the MPIB imaging facility for help in data representation, Petr Cejka, Jörg Renkawitz and members of the Pfander lab for stimulating discussion and critical reading of the manuscript. This work was supported by the German Research Council (DFG; project grant PF794/1–1, PF794/3–1 to B.P.) and the Max Planck Society (to B.P.), by the Danish Agency for Science, Technology and Innovation (DFF), the Danish National Research Foundation (DNRF115), and the Villum Foundation (to M.L.). S.C.S.B. was supported by a Chemiefonds stipend of the FCI.

## Author contributions

S.C.S.B. and B.P. conceived and designed research. M.L. performed the microscopy experiments of Fig. 3d–h and Supplementary Fig. 6B, C, D, E, F, H, I. All other experiments were performed by S.C.S.B. All authors analysed the data. B.P. wrote the paper, and S.C.S.B. and ML contributed to writing.

## Additional information

**Competing interests:** The authors declare no competing interests.

