## [Peer Review File · Nature Communications]

Reviewers' comments:

Reviewer #1 (Remarks to the Author):

The manuscript from Pfander and colleagues addresses the question of how cells sense levels of DNA damage and how that information impacts both local and global DNA damage checkpoint outputs. Using enzyme controlled site specific DSB formation in budding yeast along with various mutant backgrounds, the authors modulate the extent of end resection and the amount of ssDNA generation. Interestingly, phosphorylation of the histone H2A (a local signal) was largely independent of the amount of ssDNA present. By contrast, Rad53 kinase activation was highly dependent on the ssDNA levels. The ssDNA-dependent recruitment of the Mec1-Ddc2 kinase complex is established to regulate Rad53 activation, and the authors add to this by showing that the 9-1-1 complex also quantitatively accumulates in a manner dependent on the amount of ssDNA present. Fusion proteins designed to enhance 9-1-1 dependent signaling facilitated activation of Rad53, despite reducing ssDNA production and Mec1-Ddc2 recruitment. Together the results suggest that the global checkpoint signal is controlled not only by the amount of Mec1-Ddc2 recruited but also through the accumulation of 9-1-1 complexes at damage sites. Overall the paper is well written and the data are clear and convincing. A few issues to address include the following:

The paper focuses on spatial control of DNA damage signaling, contrasting local and global events. Several laboratories have also reported temporal control of checkpoint signaling, with the initial response centered on checkpoint activation and prevention of inappropriate resection, while later resection proceeds, enabling repair. It would be appropriate for the authors to frame their model both in terms of spatial and temporal regulation of the response.

The quality of the Rad53 blots is variable throughout the paper. In different experiments the same conditions lead to very different results in terms of the amount of Rad53 present (eg. Fig 1B vs Fig 1E; or Fig 4D vs. Fig S6B), suggesting uneven loading. It would be appropriate to include a loading control like Ponceau staining for each western blot.

The authors use a fluorescence imaging approach to quantify the amount of Rpa and 911 recruited to damage sites. They indicate that the results are normalized based on Rad52 abundance but I don't see a citation or other information about the absolute amount of Rad52 in cells. Also have the authors tested their method in the *sgs1/exo1* mutant or other backgrounds with altered resection? It seems that would be important evidence that the signal really is proportional to ssDNA levels.

The authors use H2A mutants that cannot be phosphorylated on page 6 of the text. Are these mutants equivalent to the one originally reported and characterized by Foster and Downs? It would be appropriate to cite that earlier work. The authors show that H2A phosphorylation is similar in WT and either *hta1* or *hta2* truncation mutants. Have the authors shown that H2A phosphorylation is ablated in the double mutant to confirm specificity in their assay?

The authors show that break-induced H2A phosphorylation appears normal in a *ddc1*, *dpb11*, *dna2* mutant that fails to activate Mec1. The authors attempt to rule out the possibility that this H2A phosphorylation is due to aberrant activation of Tel1. They analyze H2A in *tel1* mutants, but only do so in the single mutants for *ddc1*, *dpb11*, or *dna2*. The authors should test the triple mutant in the *tel1* deficient background to be sure that Tel1 doesn't mediate H2A phosphorylation when all three Mec1 activators are disrupted.

The discussion would benefit from comment on the evolutionary conservation of the mechanism proposed. In higher eukaryotes there is an additional ATR activator, ETAA1, that is recruited to RPA coated ssDNA and activates ATR independently of 911. It would be interesting to hear the author's thoughts on how this mechanism relates to the 911 based Mec1 activation at ssDNA

proposed in the current study.

In the opinion of this reviewer, the title is too broad in scope. A more specific title would be helpful to direct readers to this study. One of many possibilities would be something along the lines of "Quantitative sensing of ssDNA accumulation by DNA damage checkpoint mediators"

There are a few minor errors and omissions:

Line 70 'boarder' should read 'border'

Line 182, 'Fig. 2A' should be 'Fig. 2B'

The figure legend for Fig. S4 would benefit from a brief note explaining why the *sml1* deletion is used, for readers who are not familiar with the background on *mec1* mutants.

Reviewer #2 (Remarks to the Author):

This manuscript analyzes the effect of ssDNA in triggering phosphorylation of two major checkpoint targets: Rad53 and H2A. They find that Rad53 is activated in a manner that correlates with the amount of ssDNA, whereas the signal that leads to H2A phosphorylation does not respond to the amount of ssDNA. They propose the existence of a global and local checkpoint signaling. Finally, they propose 9-1-1 as ssDNA sensor.

While these findings are potentially interesting, I am not convinced that these analyses offer sufficiently conclusive insights to support this phenomenon. First, the authors use two different assays to test Rad53 and H2A phosphorylation (mobility shift for Rad53 and ChIP for H2A with phosphospecific antibodies) that differ in sensitivity. Furthermore, their results can be simply explained if Tel1 is activated in *exo1 sgs1* mutant. The authors show that deletion of TEL1 in *exo1 sgs1* mutant does not reduce H2A phosphorylation (Fig 2B). However, in Figure S4, the authors show that H2A and Rad53 are still phosphorylated in *exo1 sgs1 mec1* cells (by Tel1?) and this is in contrast with the experiment shown in Fig 2B that shows that deletion of Tel1 has no effect. Furthermore, in figure 1A, the authors show that Ddc2 (and I guess Mec1) binding at the HO-induced DSB in *exo1 sgs1* cells is detectable only very close to the DSB end, whereas H2A phosphorylation is extended for 25 kb, prompting me to ask which is the kinase that phosphorylates H2A under this condition. These results are in contrast to Shroff et al, 2004 (the paper is not mentioned), in which it was shown that H2A phosphorylation in response to an HO-induced DSB in G1 (no resection as in *exo1 sgs1* cells) is completely dependent on Tel1. How is H2A and Rad53 phosphorylation when the break is generated in *mec1* and *tel1* mutants arrested in G1? I think that a much deeper understanding of the role of Tel1 in these phosphorylation events is critical to support the model.

I also miss the point made by the authors that the 9-1-1 acts as ssDNA sensor (see uncoupling experiment with hyperactivated 9-1-1). Rad53 hyperactivation is due to an increased Rad9 binding at the DSB. Why do the authors consider 9-1-1 and not Rad9 as sensor?

Reviewer #3 (Remarks to the Author):

The authors use Gal-induced DSB in the genome, which by resection generates ssDNA stretches, to quantify the response of Mec1 checkpoint. They compared the up regulation of checkpoint related proteins and the phosphorylation level of histone H2A and Rad53. They concluded that H2A phosphorylation is independent of the amount of ssDNA generated while Rad53 is thus defining rad53 as the target for a global response. It is an interesting approach to look at the checkpoint response and Mec1 targets.

1-Throughout the paper, it was not clear which strains were used for each set of experiments. It

might be beneficial to specify the strain name in the figure legends.

2-WB of Rad53 phosphorylation showed inconsistency in the strength of the signal. Does this reflect the amount of rad53 produced at the specific time points or is there variation in the amount of total protein loaded in each lane? Did you target another protein to insure the amount loaded is comparable between all lanes? (1E and 4B)

3-Figure 1B, Rad53 phosphorylation stayed constant 2,3,4 hours although the RPA and Ddc2 almost doubled. How can this be explained?

4-What is the difference between Fig 1A, M phase WT, and Fig 1D M phase? If they are the same, why there's a huge delay (5 hours) until Rad53 showed higher phosphorylated band compared to 1A at 2 hours?

5-Why does the fold enrichment values of RPA in WT cells significantly differ between one experiment and another? What is the percent error in these values?

6-It is well established that PP2A dephosphorylate gamma-H2Ax to facilitate repair of DSB. Although the DSBs generated in your experiment are not repairable, how can you eliminate the possibility that the dephosphorylation might be continuously happening thus not allowing you to observe a major change in the phosphorylated H2A.

Point-by-point response (Bantele, Lisby and Pfander) – NCOMMS-18-10108-T

Reviewer 1

We were pleased by the very positive response of this reviewer, who finds that our “paper is well written and the data are clear and convincing”. We would very much like to thank the reviewer for his constructive criticism to our paper. We have addressed the points he/she raised experimentally and added many control experiments (Fig. 1A, B,E, 2D, 3B, 4B,D, S4A, G-H and Reviewer Fig.1, 2), as well as repeated and replaced Rad53 Western Blots throughout the manuscript (Fig. 1B,E, 3B, 4B,D).

Additionally, we now (i) added quantitative imaging and CHIP data on RPA and 9-1-1 foci under conditions of reduced (*exo1Δ*, *sgs1Δ*, and *exo1Δ sgs1Δ*) and enhanced (*rad9Δ*) DNA end resection addressing the question of ssDNA sensing (Fig. 3H-I), and (ii) added new experiments addressing the Mec1-activator and/or Tel1-dependence of the γ H2A signal (Fig. S4D, S5A-D).

Specific points:

(1) The paper focuses on spatial control of DNA damage signaling, contrasting local and global events. Several laboratories have also reported temporal control of checkpoint signaling, with the initial response centered on checkpoint activation and prevention of inappropriate resection, while later resection proceeds, enabling repair. It would be appropriate for the authors to frame their model both in terms of spatial and temporal regulation of the response.

We thank the reviewer for the thoughtful comment on the temporal aspects of checkpoint signaling. Notably, while a fast “checkpoint response” and a slow “repair response” is intuitive, our finding that γ H2A phosphorylation requires less resection compared to Rad53 activation would perhaps rather suggest that γ H2A phosphorylation is intrinsically quicker than phosphorylation of Rad53.

We have refrained from making a statement on temporal aspects in the first version of the manuscript, as we found it difficult to compare the kinetics of Rad53 phosphorylation as measured by mobility shift in Western Blot to γ H2A phosphorylation as measured by CHIP. Therefore, for the revised version of our manuscript, we have now included a new experiment, where we compare Rad53 association measured by CHIP with γ H2A phosphorylation measured by CHIP (new Fig. 1A). Notably, although the temporal resolution of our experiment is limited we do observe γ H2A phosphorylation being slightly quicker than Rad53 association, which correlates with the resection requirement of the two checkpoint responses. Overall, our results therefore do not support the idea of a fast “checkpoint response” followed by a slow “repair response”, consistent with single cell imaging data in mammalian cells that suggest that γ H2A formation at a DSB occurs within only a few minutes (Lukas et al. 2004, EMBO J). We therefore thank the reviewer for this important comment and we have integrated a comment on the temporal relationship of checkpoint signalling in the manuscript:

“Since the γ H2A response requires less ssDNA signal compared to Rad53 activation, one might predict that it is also faster. Even though the temporal resolution of our experiments is limited (30min until a DSB is induced in most cells), we find that the γ H2A response reached a plateau by 2h of DSB induction, while Rad53 association with the damage site increased over the 4h timecourse of the experiment (Fig. 1A).”

(2) The quality of the Rad53 blots is variable throughout the paper. In different experiments the same conditions lead to very different results in terms of the amount of Rad53 present (eg. Fig 1B vs Fig 1E; or Fig 4D vs. Fig S6B), suggesting uneven loading. It would be appropriate to include a loading control like Ponceau staining for each western blot.

We apologize for this. Indeed, due to different antibody batches used during the course of the project the Rad53 westerns in the first version of the manuscript showed different signal intensities and therefore may have looked partially different (not a loading issue). Initially we were not worried about this, because we were mainly focussed on the relation of hypo- to hyper-phosphorylated bands. However, we agree with the criticism and have now rerun *all* Rad53 westerns of the main figures and included two loading controls (unrelated antibody (Cdc48), Fig. 1B,E, 3B, 4B,D) and Ponceau stain (Reviewer Figure 1A-E).

(3) The authors use a fluorescence imaging approach to quantify the amount of Rpa and 911 recruited to damage sites. They indicate that the results are normalized based on Rad52 abundance but I don't see a citation or other information about the absolute amount of Rad52 in cells. Also have the authors tested their method in the *sgs1/exo1* mutant or other backgrounds with altered resection? It seems that would be important evidence that the signal really is proportional to ssDNA levels.

We thank the reviewer for the important suggestion of testing resection-deficient mutants in our quantitative imaging approach. Following this suggestions we have expanded our data to *exo1Δ*, *rad9Δ* (both accompanied by new Ddc1-3FLAG ChIP data) and *exo1Δ sgs1Δ* mutants (Fig. 3H-I, S6E-H). Our observations can be summarized as follows: (i) with ongoing resection there is increased association of the 9-1-1 complex into the DNA damage focus; (ii) when we slow down resection by the *exo1Δ* or the *sgs1Δ* mutant we reduce 9-1-1 association and importantly also the gain over time; (iii) when we accelerate resection by the *rad9Δ* mutant we increase 9-1-1 association and also the gain over time; (iv) in the *exo1Δ sgs1Δ* mutant there is no increase of 9-1-1 association over time, but (v) there is a high initial resection-independent association of the 9-1-1 complex in the *exo1Δ sgs1Δ* mutant. We feel that these data are best explained by a dual mode of association – a resection-dependent component, which dominates in *wild type* cells and other resection-permissive conditions (even in G1, Fig. 3E, S6A-C) – and a resection-independent component, which only becomes apparent in *exo1Δ sgs1Δ* cells. We also need to point out that in the *exo1Δ sgs1Δ* mutant we could not use our conventional pGAL-HO system (Fig. S6i), as these cells display frequent 9-1-1 containing but damage-independent foci, making it impossible to determine whether a given focus is HO-induced or not. Therefore, we have now used a system where an I-SceI cut-site is labelled by an adjacent TetO-array, which allows us to specifically follow the I-SceI induced DSB and test whether Ddc1/Rfa1 foci colocalize with the DSB (Fig. S6ii) Lisby et al. 2004, Cell).

Lastly, we have added a citation for the number of Rad52 molecules in the nucleus - Lisby et al., 2003, Nat Cell Biol.

(4) The authors use H2A mutants that cannot be phosphorylated on page 6 of the text. Are these mutants equivalent to the one originally reported and characterized by Foster and Downs? It would be appropriate to cite that earlier work. The authors show that H2A phosphorylation is similar in WT and either *hta1* or *hta2* truncation mutants. Have the authors shown that H2A phosphorylation is ablated in the double mutant to confirm specificity in their assay?

Our H2A phosphorylation site mutants are indeed highly similar to those initially characterized by Downs, Lowndes and Jackson, *Nature*, 2000, but expressed from the endogenous loci. We appreciate the importance of a specificity control and have now added in Fig. 2A and S4A an experiment including *hta1-S129STOP hta2-S129STOP* cells, which we had obtained from the same crossing as the presented single mutants. As expected, γ H2A ChIP signals were observed in WT but not *hta1-S129STOP hta2-S129STOP* double mutant cells, demonstrating specificity of our γ H2A ChIPs.

Notably, as also described in the paper, we in this experiment show γ H2A ChIP data as IP/input ratios instead of data normalized to control loci on undamaged chromatin, as we (expectedly)

find that also basal, genome-wide γ H2A levels are strongly impaired in the *hta1-S129STOP hta2-S129STOP* double mutant. We chose a similar procedure for all mutants (for example *mec1 Δ* , the Mec1 activator mutant or the mutant in γ H2A phosphatase Pph3 (Fig. 2A-D, S4A-D, S5A,C,E,F).

(5) The authors show that break-induced H2A phosphorylation appears normal in a *ddc1, dpb11, dna2* mutant that fails to activate Mec1. The authors attempt to rule out the possibility that this H2A phosphorylation is due to aberrant activation of Tel1. They analyze H2A in *tel1* mutants, but only do so in the single mutants for *ddc1*, *dpb11*, or *dna2*. The authors should test the triple mutant in the *tel1* deficient background to be sure that Tel1 doesn't mediate H2A phosphorylation when all three Mec1 activators are disrupted.

We would like to thank the reviewer for this thoughtful comment, which has prompted us to revisit the influence of Mec1 activators on γ H2A phosphorylation. Indeed we observe that γ H2A phosphorylation is affected by the presence of Mec1 activators. When measuring γ H2A phosphorylation by CHIP, we find that IP/input ratios in proximity to the DSB, as well as the background γ H2A signal before DSB induction are strongly reduced in *ddc1 Δ dna2-WYAA* cells (Fig. 2D, S5A-B), a mutant background abolishing all well-characterized Mec1 activators. This effect is very similar to what can be observed in *mec1 Δ sml1 Δ* cells, although not as pronounced as what is observed in *hta1-S129STOP hta2-S129STOP* (see Fig. 2A-B,D, S4A-C).

In *ddc1 Δ dna2-WYAA* cells, we observe only a very small damage-specific γ H2A signal around the DSB, but this appears to be Tel1-dependent, as the reviewer expected. When we followed the suggestion of the reviewer and generated the *tel1 Δ ddc1 Δ dna2-WYAA* mutant, the damage-specific γ H2A signal was entirely lost (Fig. S5A). This effect by the *tel1 Δ* mutation is highly similar to what happens if *TEL1* is deleted in the *mec1 Δ sml1 Δ* background in G1 (see new Fig. S4D). Overall, we therefore conclude that γ H2A phosphorylation is dependent on Mec1 activators, but individual activators appear to act redundantly (Fig. S5C-D). Therefore, we conclude that no single activator appears to be specific or limiting γ H2A phosphorylation.

(6) The discussion would benefit from comment on the evolutionary conservation of the mechanism proposed. In higher eukaryotes there is an additional ATR activator, ETAA1, that is recruited to RPA coated ssDNA and activates ATR independently of 911. It would be interesting to hear the author's thoughts on how this mechanism relates to the 911 based Mec1 activation at ssDNA proposed in the current study.

We fully agree with the reviewer that this is an important question. Indeed, we think that different "sensitivities" for ssDNA may be an underlying force for the evolution of different Mec1/ATR activators. Therefore, we have added this paragraph to the discussion:

"It appears highly likely that this mechanism of dual recognition of checkpoint signals is present in higher eukaryotes, too, given that the involved proteins as well as their association with single-stranded DNA are highly conserved throughout evolution. Notably, mammalian cells feature another activator for ATR (human Mec1), ETAA1, which binds to RPA as well, but is independent of the 9-1-1 complex (68-70). It therefore seems reasonable to hypothesize that ETAA1 will – similar to the 9-1-1 complex – quantitatively contribute to checkpoint signalling."

(7) In the opinion of this reviewer, the title is too broad in scope. A more specific title would be helpful to direct readers to this study. One of many possibilities would be something along the lines of "Quantitative sensing of ssDNA accumulation by DNA damage checkpoint mediators"

Yes, agreed. We have now chosen a more specific title: "Quantitative sensing and signalling of single stranded DNA during the DNA damage response".

(8) There are a few minor errors and omissions:

Line 70 'boarder' should read 'border'

Line 182, 'Fig. 2A' should be 'Fig. 2B'

The figure legend for Fig. S4 would benefit from a brief note explaining why the *sml1* deletion is used, for readers who are not familiar with the background on *mec1* mutants.

Thank you! All corrected and included the explanation.

Reviewer 2

We thank the reviewer for the constructive criticism and important considerations he/she had to our story. The criticism has also revealed points where our manuscript lacked clarity. We have addressed all points raised in full and with new experiments shown in Fig. 1A, 2B, S1E, S4D-F and Reviewer Fig. 2A-B and hope that the reviewer will agree with our conclusions.

Notably, the reviewer has also raised a general concern regarding the fact that key conclusions in the paper are relying on the comparison of two different assays (Rad53 phosphorylation measurements by western blots and γ H2A phosphorylation measurements by ChIP). We share this concern, but for general reasons and supported by new experiments outlined in (1), we are convinced that different sensitivities in the assays will not influence our conclusion.

(1) First, the authors use two different assays to test Rad53 and H2A phosphorylation (mobility shift for Rad53 and ChIP for H2A with phosphospecific antibodies) that differ in sensitivity.

Many thanks for bringing up this critical point. Mobility shift in Western Blot (Rad53) and ChIP with a phosphorylation-specific antibody (γ H2A) are clearly two assays that differ in sensitivity. ChIP is generally considered a more sensitive and quantitative read-out than Western Blot. Indeed this is also true for our system, as we can for example observe a clear induction of γ H2A signal surrounding the DSB by ChIP, while by western blot it is very difficult to observe a DSB-induced γ H2A signal over the high γ H2A background in M-phase arrested cells (Fig. S1E). Therefore, we would consider it problematic, if the more sensitive assay (γ H2A ChIP) would show changes, while the less sensitive assay (Rad53 Western Blot) would not. The opposite is true for our study: the more sensitive assay (γ H2A ChIP) is giving a similar signal independent of DNA end resection, while the less sensitive assay (Rad53 Western Blot) is giving changing signals that depend on resection. Therefore, we do think that our experimental strategy is in principle valid.

To address this point of concern further, we conducted two additional experiments. First, we performed ChIP for Rad53 with *WT* and *exo1 Δ sgs1 Δ* cells (see new panel in Fig. 1A). Consistent with our model, we observed that more Rad53 associated in the presence of more ssDNA in *WT* cells compared to the *exo1 Δ sgs1 Δ* scenario. Even though this assay measures Rad53 protein and not necessarily its phosphorylated form, we therefore note that the Rad53 response to a DSB is highly dependent on resection and the amount of ssDNA formed at the DSB in two different assays (ChIP and Western Blot).

Second, we also show a side-by-side comparison of γ H2A ChIPs and Western blots (Fig. S4D, Reviewer Fig. 2A-B). As outlined above the overall, genome-wide background of γ H2A is relatively high in M-arrested cells, therefore this experiment was performed in G1-arrested cells. Notably, however, the DSB-induced γ H2A signal in G1 cells is similar in ChIP and western blot (Fig. S4D, Reviewer Fig. 2A-B). Overall, these data therefore support our conclusion that γ H2A is sensitive to much lower amounts of ssDNA signal compared to Rad53.

(2) Furthermore, their results can be simply explained if Tel1 is activated in *exo1 sgs1* mutant. The authors show that deletion of *TEL1* in *exo1 sgs1* mutant does not reduce H2A phosphorylation (Fig 2B). However, in Figure S4, the authors show that H2A and Rad53 are still phosphorylated in *exo1 sgs1 mec1* cells (by Tel1?) and this is in contrast with the experiment shown in Fig 2B that shows that deletion of *Tel1* has no effect.

Compensation by Tel1 was also the first idea that came to our mind, as previous work suggested that a handover from Tel1 signalling to Mec1 signalling occur upon resection initiation (Clerici et al., *EMBO J*, 2014). In theory, therefore, one might expect that by inhibiting DNA end resection this handover will be blocked. In Figure S4E-F, we indeed observed support for slightly stronger Tel1 activity in resection-impaired cells specifically after *phleomycin* treatment. It is however not clear how much this effect might contribute to the overall γ H2A signal (note the rather minor effect in the side-by-side γ H2A western, which we now added as Fig. S4E).

Already in the previous version of our paper, we therefore tested the handover model by using a deletion of *TEL1* and observed that after *DSB induction with HO*, γ H2A signals were Tel1-independent also in *exo1 Δ sgs1 Δ* cells (Fig. 2C). In order to strengthen this point, we now generated a *mec1 Δ sml1 Δ exo1 Δ sgs1 Δ* strain and compared DSB-induced γ H2A signals in *mec1 Δ sml1 Δ* and *mec1 Δ sml1 Δ exo1 Δ sgs1 Δ* cells. Notably, both strains gave essentially baseline signals for γ H2A (Fig. 2B) and we therefore conclude that HO-DSB-induced γ H2A phosphorylation is by large Mec1-dependent in M phase-arrested *WT* and *exo1 Δ sgs1 Δ* cells. We would like to note that our data does not contradict the “handover model”, it is entirely possible that *exo1 Δ sgs1 Δ* cells arrests resection at a point, where the Tel1/Mec1 handover has already occurred.

(3) Furthermore, in figure 1A, the authors show that Ddc2 (and I guess Mec1) binding at the HO-induced DSB in *exo1 sgs1* cells is detectable only very close to the DSB end, whereas H2A phosphorylation is extended for 25 kb, prompting me to ask which is the kinase that phosphorylates H2A under this condition.

As outlined in (2), we now included in Fig. 2B an experiment that shows that the entire γ H2A phosphorylation signal is abolished in *mec1 Δ sml1 Δ* and *mec1 Δ sml1 Δ exo1 Δ sgs1 Δ* cells (with *TEL1* deletion not showing a significant effect (Fig. 2C)). Therefore, we conclude that γ H2A phosphorylation at an HO-induced DSB is Mec1-dependent.

As such Mec1 is likely the H2A kinase under these conditions and indeed this raises the important point, how a kinase that is localized to a specific site of the genome can spread a phosphorylation signal over >100 kb (or even 1 Mb in mammalian cells). We discuss possible mechanisms in the paper. A mechanism depending on contacts/interactions between different sites of the chromosome and thereby chromosome architecture seems most straightforward. Notably, such a mechanism would also offer a plausible explanation for the unique behaviour of H2A phosphorylation and could also explain why the amount of Mec1 kinase associated with the DSB is not limiting to H2A phosphorylation.

(4) These results are in contrast to Shroff et al, 2004 (the paper is not mentioned), in which it was shown that H2A phosphorylation in response to an HO-induced DSB in G1 (no resection as in *exo1 sgs1* cells) is completely dependent on Tel1. How is H2A and Rad53 phosphorylation when the break is generated in *mec1* and *tel1* mutants arrested in G1? I think that a much deeper understanding of the role of Tel1 in these phosphorylation events is critical to support the model.

We thank the reviewer for pointing us to the fact that we did not reference the Shroff et al. (2004) paper. This was indeed a milestone paper characterizing γ H2A by ChIP, which needs to be (and is now) cited in our study. Using a semi-quantitative set-up, this study also demonstrated a high degree of Tel1-dependence of the γ H2A signal in G1-arrested cells (Shroff et al., *Curr Biol*, 2004, Fig. 4C). We revisited this question, as the initial conclusion of the Shroff paper hinged on the measurement of a single locus and timepoint and relied on quantification of band-intensities after multiplex PCR rather than qPCR. For the revised version of our paper, we added a γ H2A ChIP experiment comparing *WT*, *mec1 Δ* , *tel1 Δ* and *mec1 Δ tel1 Δ* cells arrested in G1 (Fig. S4D). In this experiment, we observe a partial reduction of the γ H2A signal in cells lacking Tel1. Notably, in our experimental setup this reduction of the γ H2A signal is similar, but actually weaker compared to the reduction in cells lacking Mec1. In *mec1 Δ tel1 Δ* double mutant cells the γ H2A ChIP signal is entirely lost. Therefore, if we compare the two resection-deficient scenarios (Fig. 2B-C), this suggests that in G1 cells the contribution of Tel1 to γ H2A phosphorylation is stronger, compared to M phase arrested *exo1 Δ sgs1 Δ* cells. As such we do not see a contradiction with the Shroff et al paper and importantly the new data further supports the major conclusion of our paper that relatively little resection and therefore very little Mec1-Ddc2 association is sufficient to elicit a full-blown γ H2A response. Given that very little resection is also sufficient to switch from Tel1 to Mec1, it is difficult to ascertain whether Tel1 alone is also able to elicit a full-blown

γ H2A response, but we think it is very likely that very few molecules of Tel1 would also be sufficient.

(5) I also miss the point made by the authors that the 9-1-1 acts as ssDNA sensor (see uncoupling experiment with hyperactivated 9-1-1). Rad53 hyperactivation is due to an increased Rad9 binding at the DSB. Why do the authors consider 9-1-1 and not Rad9 as sensor?

We thank the reviewer for his/her comment on 9-1-1 vs Rad9 as putative sensor in checkpoint signalling. We apologize if the first version of our manuscript was not sufficiently clear. Indeed, we find that all proteins that associate with DNA damage sites via 9-1-1 (9-1-1 itself, Dpb11 and Rad9) are strongly influenced by resection. Therefore we reason that a function of the entire 9-1-1 axis is to quantitatively transduce this signal. We simply focused our efforts on the 9-1-1 complex, because it (i) is furthest upstream (Fig. 3A-B) and (ii) associates independently of Mec1-Ddc2 (Kondo et al., *Science*, 2001; Melo et al., *Genes Dev*, 2001; Zou et al., *PNAS*, 2002). Dpb11- and Rad9- association depends on the 9-1-1 complex, but it also depends on Mec1-Ddc2 (Fig. 3A-B, Puddu et al., *Mol Cell Biol*, 2008). Therefore, in case of Rad9 it is impossible to discriminate, whether the observed resection-dependence originates in 9-1-1 associating resection-dependently,, Mec1-Ddc2 associating resection-dependently or both. How the association of 9-1-1 and Mec1-Ddc2 is quantitatively transduced to downstream factors is a very interesting question, which however goes beyond the scope of our manuscript.

Reviewer 3

We thank the reviewer for agreeing with the validity of our approach and are very pleased to hear he/she finds our paper interesting. We would also like to thank the reviewer for the constructive criticism. The reviewer raised important points regarding the analysis of our data as well as to the interpretation. We now include in the new version of our paper substantial additional experiments and additional analysis (Fig. 1A,B,E, 3B, 4B,D, S1A, S5E-F, Reviewer Fig. 1A-E which we hope will address the criticism in full.

(1) Throughout the paper, it was not clear which strains were used for each set of experiments. It might be beneficial to specify the strain name in the figure legends.

Agreed! As strain names in the figure legends are incompatible with the journals space restrictions, we have now included a new column in Table 1, indicating which experiments the respective strains were used for, and hope that this will help and direct the reader to easily understand which strains were used in which experiments.

(2) WB of Rad53 phosphorylation showed inconsistency in the strength of the signal. Does this reflect the amount of rad53 produced at the specific time points or is there variation in the amount of total protein loaded in each lane? Did you target another protein to insure the amount loaded is comparable between all lanes? (1E and 4B)

We apologize for Rad53 western blots having different signal strength in the first version of the manuscript. Different Western Blots were done over a 4-year period and different batches of Rad53 antibody were used, thus the different signal strength and background. We have now repeated *all* Rad53 western blots in the main figures (Fig. 1B,E, 3B and 4B, D). As suggested, for each we have also included loading controls using an antibody against an unrelated protein (Cdc48), and additionally provide you with the Ponceau staining of the membranes (Reviewer Fig. 1A-E).

(3) Figure 1B, Rad53 phosphorylation stayed constant 2,3,4 hours although the RPA and Ddc2 almost doubled. How can this be explained?

We thank the reviewer for pointing us towards the kinetics of Rad53 activation, which was somewhat neglected in the first version of our paper. We have now included an entire paragraph comparing the kinetics of Rad53 phosphorylation to those of γ H2A phosphorylation. Indeed, we observed repeatedly (see below) that Rad53 phosphorylation in the 2,3,4 h samples appears relatively similar, even though resection is ongoing. To have a more quantitative and comparable read-out related to Rad53 activation, we tested Rad53 association to the DSB (which is thought to be a prerequisite of phosphorylation by Rad53 ChIP-qPCR. Notably, we found that the Rad53 ChIP signal increased over time and apparently mirrors the RPA ChIP signal (Fig. 1A). Overall, our data therefore suggest that Rad53 associates with the DSB – and likely gets activated – in a manner that is highly correlated with the amount of ssDNA and RPA. Given however, that the amount of phosphorylated Rad53 reaches a plateau, we reason that it is likely to be underlying additional influences, for example by phosphatases, leading to saturation. In the revised version of the manuscript, we point to a non-linear relationship in the discussion.

(4) What is the difference between Fig 1A, M phase WT, and Fig 1D M phase? If they are the same, why there's a huge delay (5 hours) until Rad53 showed higher phosphorylated band compared to 1A at 2 hours?

Looking at the old westerns from the first version of the paper, we can appreciate why the reviewer thinks the kinetics of Rad53 activation might have been different in Fig. 1B and Fig. 1E, but we attribute this seeming difference to the exposures with different batches of Rad53

antibody. When we checked longer exposures it was very clear that also in Fig. 1E there is efficient Rad53 phosphorylation 2h after HO induction. This is also clear in our standardized repetitions of the Rad53 westerns (new Fig. 1B, 1E, 3B, 4B, D), and we are very confident this is an accurate description of the behaviour of WT cells, since we have observed this now in >10 repetitions of this experiment.

(5) Why does the fold enrichment values of RPA in WT cells significantly differ between one experiment and another? What is the percent error in these values?

This is an important technical concern. While spreading of the ChIP signals as well as the overall shape of the ChIP curve are highly reproducible, we find that the overall “fold enrichment” values fluctuate significantly between experiments. In contrast, within one experiment the “fold enrichment” values are highly similar. In order to address this concern, we generated three independent break induction samples (biological replicates), prepared extracts side-by-side and measured RPA as well as γ H2A ChIP signals and confirmed them to be highly reproducible within the experiment (error bars represent the standard deviation, which ranges between 5-20%) (Fig. S1A). In comparison, when we plotted all RPA and γ H2A ChIP signals from all individual WT cells obtained in the course of this study and overlaid them, we find differences in the fold enrichment, even though there is a very good correlation in how far the signals spread (Fig. S1A). Therefore, in the design of the study, we have chosen to only compare data (and present them in one panel) generated in the same experiment, but have also reproduced them in one or more entirely independent biological replicate, which always confirmed the trends observed in the data included in the manuscript.

(6) It is well established that PP2A dephosphorylate gamma-H2Ax to facilitate repair of DSB. Although the DSBs generated in your experiment are not repairable, how can you eliminate the possibility that the dephosphorylation might be continuously happening thus not allowing you to observe a major change in the phosphorylated H2A.

In budding yeast, γ H2A is largely removed by PP4 (Pph3; Krogan lab: Keogh et al., *Nature*, 2006; Haber lab: Kim et al., *MCB* 2011) with PP2C playing a very minor role. PP2A in contrast seems to play a more important role in mammalian cells (Chowdhury et al., *Mol Cell*, 2005). In response to this comment, we therefore tested a *pph3 Δ* mutant and first of all found that under this condition γ H2A was abundantly present throughout chromatin even in the absence of exogenous damage (Fig. S5E-F see elevated signals at 0h). However, upon DSB induction, we observed very similar γ H2A signals in *WT* and *pph3 Δ* cells, suggesting that dephosphorylation by PP4 is not limiting γ H2A during the immediate response to a DSB.

Indeed, this finding is consistent with the idea of γ H2A being a relatively stable phosphorylation mark that is not rapidly turned over by phosphatases. We thank the reviewer for this idea, we added the data to Fig. S4, as now we can also rule out that γ H2A phosphorylation is essentially phosphatase-controlled.

A

B

Reviewer Figure Legend:

Reviewer Figure 1

(A-E) Additional loading controls for Western Blot-based Rad53 activation experiments shown in **(A)** Fig. 1B **(B)** Fig. 1E, **(C)** Fig. 3B, **(D)** Fig. 4B, **(E)** Fig. 4D. The Western blot images are the same as displayed in the main figures of this manuscript, here an additional Ponceau staining to control for equal amounts of extract in each lane is shown.

Reviewer Figure 2

(A) Both Mec1 and Tel1 contribute to γ H2A phosphorylation in G1. IP/input ratios of γ H2A ChIP after indicated times of DSB induction in G1 phase. WT cells were compared to *tel1 Δ* , *mec1 Δ* and *mec1 Δ tel1 Δ* double mutant cells (This experiment is shown in Figure S4D).

(B) γ H2A analysis by Western blot yields similar results as ChIP-qPCR analysis. Western Blot against γ H2A in cells arrested in G1 and after 4h of DSB induction. WT cells were compared to *tel1 Δ* , *mec1 Δ sml1 Δ* , and *mec1 Δ sml1 Δ tel1 Δ* mutant cells. Samples were taken from cultures of the γ H2A ChIP-qPCR experiment shown in Fig. S4D.

REVIEWERS' COMMENTS:

Reviewer #1 (Remarks to the Author):

The authors have thoroughly responded to the prior reviews by improving the quality of the figures, adding important controls, and clarifying some of the interpretations. Only a few minor issues remain.

1. In responding to reviewer 2, point 1, the authors make the case that they see similar results for H2A phosphorylation by ChIP and western blotting. This conclusion is based on data that are provided for the reviewers (ChIP in reviewer Fig 2A; western blot in reviewer Fig 2B). The ChIP data are also included in the paper (Fig. S4D) but the western blot is not. It would be appropriate to include Reviewer Fig 2B (the western) in the supplementary data, alongside Fig. S4D.

2. A minor stylistic point relates to figures 4B and 4D. Instead using the conventional '+' symbol to indicate which samples included expression of the fusions, the figure uses dots. Presumably the authors have a reason that I'm missing for labeling in this manner but I'd consider '+' to be more intuitive and clearer.

Reviewer #2 (Remarks to the Author):

The authors have addressed all my concerns and the manuscript has been improved. I agree with the publication.

Reviewer #3 (Remarks to the Author):

The authors reply to my concerns are sufficient for this manuscript.

Future work might be to understand whether the non-quantitative nature of the local checkpoint signalling of H2A is M phase specific or general and whether it holds true in response to various kinds of damage other than DSB.

Please check the legend for Figure S6 (C) "before addition". Is that correct?

REVIEWERS' COMMENTS:

Reviewer #1 (Remarks to the Author):

The authors have thoroughly responded to the prior reviews by improving the quality of the figures, adding important controls, and clarifying some of the interpretations. Only a few minor issues remain.

1. In responding to reviewer 2, point 1, the authors make the case that they see similar results for H2A phosphorylation by ChIP and western blotting. This conclusion is based on data that are provided for the reviewers (ChIP in reviewer Fig 2A; western blot in reviewer Fig 2B). The ChIP data are also included in the paper (Fig. S4D) but the western blot is not. It would be appropriate to include Reviewer Fig 2B (the western) in the supplementary data, alongside Fig. S4D.

The Western blot earlier presented in Reviewer Fig. 2B was included in the Supplementary Fig. 4, panel E.

2. A minor stylistic point relates to figures 4B and 4D. Instead using the conventional '+' symbol to indicate which samples included expression of the fusions, the figure uses dots. Presumably the authors have a reason that I'm missing for labeling in this manner but I'd consider '+' to be more intuitive and clearer.

In order to prevent visual confusion, we decided to use the more simplistic shape of a dot instead of a plus. We changed the dots to plusses now.

Reviewer #2 (Remarks to the Author):

The authors have addressed all my concerns and the manuscript has been improved. I agree with the publication.

Reviewer #3 (Remarks to the Author):

The authors reply to my concerns are sufficient for this manuscript. Future work might be to understand whether the non-quantitative nature of the local checkpoint signalling of H2A is M phase specific or general and whether it holds true in response to various kinds of damage other than DSB.

Please check the legend for Figure S6 (C) "before addition". Is that correct?

The statement "before addition" is correct, we nonetheless re-formulated the sentence so that it becomes more clear at which time points the FACS samples were taken.